# Self-supervised Learning of Echocardiographic Video Representations via Online Cluster Distillation

**Divyanshu Mishra**[1]     **Mohammadreza Salehi**[4]     **Pramit Saha**[1]     **Olga Patey**[2]
**Aris T. Papageorghiou**[2]     **Yuki M. Asano**[3]     **J. Alison Noble**[1]

[1]Department of Engineering Science, University of Oxford
[2]Nuffield Department of Women's and Reproductive Health, University of Oxford
[3]Fundamental AI Lab, University of Technology Nuremberg
[4] University of Amsterdam
`divyanshu.mishra@eng.ox.ac.uk`

## Abstract

Self-supervised learning (SSL) has achieved major advances in natural images and video understanding, but challenges remain in domains like echocardiography (heart ultrasound) due to subtle anatomical structures, complex temporal dynamics, and the current lack of domain-specific pre-trained models. Existing SSL approaches such as contrastive, masked modeling, and clustering-based methods struggle with high intersample similarity, sensitivity to low PSNR inputs common in ultrasound, or aggressive augmentations that distort clinically relevant features. We present DISCOVR (Distilled Image Supervision for Cross Modal Video Representation), a self-supervised dual branch framework for cardiac ultrasound video representation learning. DISCOVR combines a clustering-based video encoder that models temporal dynamics with an online image encoder that extracts fine-grained spatial semantics. These branches are connected through a semantic cluster distillation loss that transfers anatomical knowledge from the evolving image encoder to the video encoder, enabling temporally coherent representations enriched with fine-grained semantic understanding.Evaluated on six echocardiography datasets spanning fetal, pediatric, and adult populations, DISCOVR outperforms both specialized video anomaly detection methods and state-of-the-art video-SSL baselines in zero-shot and linear probing setups,achieving superior segmentation transfer and strong downstream performance on clinically relevant tasks such as LVEF prediction. Code available at: https://github.com/mdivyanshu97/DISCOVR

## 1   Introduction

Modeling dynamic content in video data presents significant challenges due to complex spatio-temporal relationships, high redundancy between frames, and the need to capture both short- and long-range temporal dependencies [41, 32]. Echocardiography (heart or cardiac ultrasound) exemplifies these video understanding challenges [23, 22]. With high frame rates (30–80 fps) [21], complex anatomical motion, and variability in image appearance caused by speckle, shadowing artifacts, and ultrasound probe variability [18], automated echocardiography analysis requires sophisticated temporal modeling approaches. The information density in these videos is high, where features critical for diagnosis may appear as subtle variations in wall motion, valve function, or blood flow patterns that manifest only when viewed dynamically across multiple frames. Moreover, the appearance of the heart can change drastically across different cardiac views, patient populations, and imaging equipment. Developing robust video SSL models for comprehensive video understanding in echocardiography faces additional obstacles due to data limitations. Expert annotations are costly, labor-intensive, and if based on real-world hospital data often incomplete, capturing only specific

39th Conference on Neural Information Processing Systems (NeurIPS 2025).

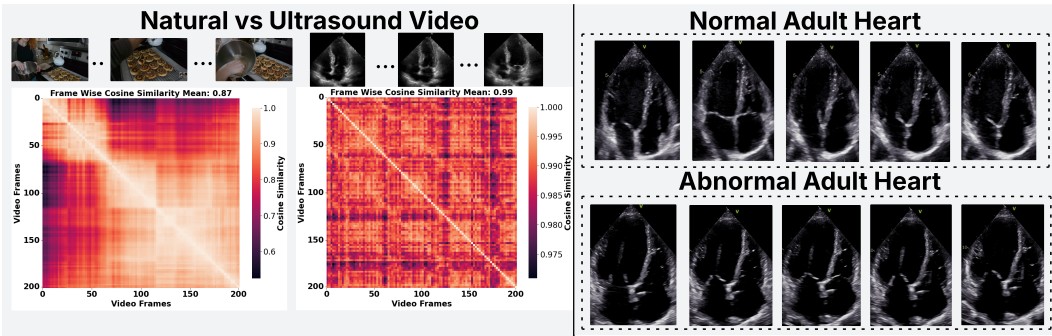

Figure 1: Figure (left) compares two fine-grained videos: a natural scene of a person baking (left) and an adult fetal heart ultrasound (right). The frame-level cosine similarity matrix, computed using a pretrained VideoMAE model, shows that ultrasound frames are highly similar (mean=0.99), with only minor local variations. This highlights the difficulty in distinguishing individual frames in such medical videos. Figure (right) compares normal and abnormal adult echocardiograms that appear nearly identical. However, on close inspection, it is revealed that the abnormal heart shows severe biventricular systolic dysfunction and a dilated, globular left ventricle, underscoring the subtlety of cardiac defects and the need for fine-grained structural analysis.

aspects of the rich information contained in these videos. This scarcity of labeled data motivates SSL approaches that can leverage abundant unlabeled echocardiograms for model development [32, 24, 6].

Several SSL frameworks have been proposed for learning meaningful video representations, each with particular limitations in the echocardiography context. Masked video modeling methods [35, 10, 9] tend to focus on reconstructing low-level image features like textures or edges, limiting their ability to capture high-level semantic information critical for clinical interpretation. This is especially problematic for ultrasound, which inherently exhibits a low signal-to-noise ratio (SNR), making approaches that rely on low-level pixel representations ineffective. Contrastive learning methods [13, 27] struggle due to high inter-sample similarity and limited effective augmentations, making it difficult to construct informative positive and negative pairs, often leading to representation collapse. Clustering-based SSL methods have demonstrated strong semantic learning through self-distillation but rely heavily on aggressive augmentations that risk disrupting essential anatomical details required for fine-grained understanding.

To address these limitations, we propose DISCOVR (*Distilled Image Supervision for Cross-Modal Video Representation*), a dual branch SSL framework tailored for echocardiography that jointly captures temporal dynamics and fine-grained semantic structure. The video encoder is trained to model temporal features using a clustering-based objective applied to masked video tokens, while an online image encoder separately learns spatially rich and anatomically meaningful representations from masked image views. To bridge the gap between spatial and temporal learning, we introduce a semantic cluster distillation loss that transfers knowledge from the evolving image encoder to the video encoder through semantic cluster alignment. This enables the video encoder to embed fine-grained semantic detail into its temporally coherent representations, without relying on pretrained models or heavy augmentations.

We extensively evaluate DISCOVR on six echocardiography datasets that span fetal, pediatric, and adult populations, covering anomaly detection, classification (linear probing and zero-shot transfer), and segmentation tasks. DISCOVR consistently outperforms prior self-supervised and anomaly detection methods. It achieves an average F1 improvement of 3.4% for anomaly detection, a 2.4% gain in linear probing, and a 1.5% increase in balanced accuracy under zero shot evaluation. For segmentation, DISCOVR delivers a 3.1% relative improvement in Dice score (from 81.9 to 84.4), despite using a simple segmentation head compared to more complex baseline architectures. These results demonstrate that integrating spatial semantics with temporal dynamics through cross-modal distillation yields robust and generalizable cardiac ultrasound video representations.

Overall, our contibutions are as follows:

- We develop an SSL method that jointly models temporal dynamics and spatial semantics by integrating video self-distillation with an evolving semantic image encoder, without labels, pretrained models, or augmentations.

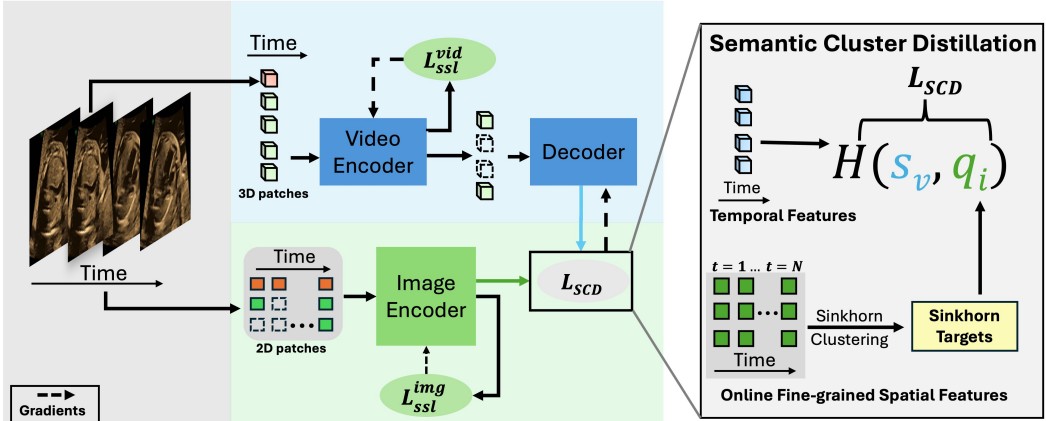

Figure 2: **Overview of the DISCOVR framework.** An input video is tokenized into 3D patches for the video branch and per-frame 2D patches for the image branch. Both encoders perform masked self-distillation. Masked video tokens are reconstructed by the video decoder, and dense semantic features are extracted from the image encoder. The $\mathcal{L}_{\text{SCD}}$ loss then aligns these outputs, distilling fine-grained spatial semantics into the video representation to produce rich spatio-temporal features.

- We introduce a novel online semantic distillation loss that continually transfers anatomical knowledge from the evolving image encoder to the video encoder, enriching its temporal representations with fine-grained spatial semantics to better capture clinically relevant spatio-temporal patterns in echocardiography.

- DISCOVR is, to our knowledge, the most comprehensive self-supervised video representation model for echocardiography to date. Trained solely on normal videos, it models healthy heart dynamics and detects pathology as deviations, eliminating the need for labeled abnormal cases. Evaluated across six datasets spanning fetal, pediatric, and adult cohorts, DISCOVR demonstrates strong generalization in zero-shot classification, linear probing, anomaly detection, and segmentation, and achieves state-of-the-art performance on downstream cardiac function estimation (LVEF prediction), making it a versatile backbone for ultrasound analysis.

## 2 Related Work

Self-supervised learning (SSL) aims to learn feature extractors directly from raw data by solving an intrinsic task using supervision signals derived from the data itself, eliminating the need for manual labels. Early image-based SSL relied on handcrafted pretext tasks such as solving jigsaw puzzles [26], predicting rotations [11], or colorizing grayscale inputs [45]. Recent methods have shifted towards instance discrimination via contrastive learning [42, 13, 5]. To understand how these ideas extend to video and medical domains, we review the most relevant self-supervised methods in both areas, highlighting shared limitations and how DISCOVR addresses them.

**Video Self-Supervised Learning.** Extending SSL to video introduces additional temporal complexity, inspiring tasks such as frame order prediction [25, 43], spatio-temporal jigsaws [16], and playback pace prediction [3, 38]. Recently, masked video modeling has become the dominant approach: VideoMAE [35] reconstructs raw pixels from masked tubelets using a ViT backbone. MGMAE [14] predicts optical flow to enhance temporal modeling, and motion-aware masking [9] highlights dynamic regions. SIGMA [33] replaces pixel-level targets with Sinkhorn-regularized cluster assignments, encouraging learning of semantic features. Yet, these approaches often rely on frozen teachers, handcrafted objectives, or sensitive clustering parameters. DISCOVR addresses these issues by introducing video self-distillation with evolving semantic guidance from an image encoder, aligning fine-grained spatial and temporal features to produce coherent, high-level video representations, without external supervision, handcrafted tasks, or modality-specific assumptions.

**Self-Supervised Pretraining for Medical Videos.** Given the limited availability of annotated data, several works have adapted video SSL techniques to medical domains. Jiao et al. [15] explored

frame order and transformation prediction for fetal ultrasound. EchoFlow [31] generated synthetic echocardiograms via adversarial VAEs and latent flow. Although effective in context, these methods inherit key limitations from natural video SSL, including reliance on frozen teachers, hand-crafted objectives, and sensitive clustering parameters. In addition, they adopt design choices tailored to natural images, such as short clip lengths and the lack of mechanisms for capturing fine-grained spatial cues, both of which are inadequate for clinical video analysis, where longer temporal context and detailed spatial reasoning are critical. In contrast, DISCOVR uses long (64-frame) clips and introduces dynamic semantic guidance from an evolving image encoder, enabling the video backbone to learn rich, fine-grained spatio-temporal representations without reliance on pretrained models or handcrafted supervision.

## 3 Methodology

The modelling of echocardiography video-based tasks poses unique challenges, as models must simultaneously detect fine-grained anatomical details, such as subtle septal defects, and accurately track how these features evolve throughout the cardiac cycle to reliably identify anomalies. We propose a unified self-supervised framework addressing these aspects without relying on labelled data or external pretrained models. Our method integrates three complementary techniques: (1) video self-distillation to capture global cardiac motion, (2) online spatial guidance to learn fine-grained structural information, and (3) semantic cluster distillation (SCD) loss to transfer fine-grained semantic knowledge from the evolving image encoder to the video model.

### 3.1 Video Self-Distillation

To capture how cardiac structures evolve throughout the cardiac cycle, it is essential to learn spatio-temporal representations from echocardiography videos. We propose a video-level self-distillation framework based on a student-teacher architecture with Vision Transformer (ViT)-based encoders (Fig. 2) that models temporal dynamics and improves understanding of global heart motion. Given a video input $v$, we partition it into non-overlapping 3D space-time patches (tube tokens), and prepend a learnable class (CLS) token, resulting in a sequence $x_0, x_1, \ldots, x_N$, where $x_0$ is the CLS token.

The teacher encoder $E_{\theta_t}$ processes the complete, unmasked video to produce a global representation, whereas the student encoder $E_{\theta_s}$ processes multiple masked variants $v_{\mathcal{M}1}, \ldots, v_{\mathcal{M}_M}$, each applying distinct random space-time masks to enforce inference of missing content.

Both encoders output a global video representation via the CLS token:

$$z_t = E_{\theta_t}(v)[0], \quad z_s^{(m)} = E_{\theta_s}(v_{\mathcal{M}_m})[0]. \tag{1}$$

The teacher parameters are updated using an exponential moving average (EMA) of the student parameters:

$$\theta_t \leftarrow \lambda\theta_t + (1 - \lambda)\theta_s, \quad \lambda \in [0, 1). \tag{2}$$

These CLS embeddings are subsequently mapped through linear projection heads characterized by learnable weight matrices $W_t$ (teacher) and $W_s$ (student). The resulting embeddings are transformed into probability distributions via temperature-scaled softmax operations:

$$P_t = \text{softmax}\left(\frac{W_t z_t}{\tau_t}\right), \quad P_s^{(m)} = \text{softmax}\left(\frac{W_s z_s^{(m)}}{\tau_s}\right), \tag{3}$$

where $\tau_t$ and $\tau_s$ are temperature parameters for the teacher and student, respectively.

We align these probability distributions using the cross-entropy loss:

$$\mathcal{L}_{\text{ssl}}^{vid} = \frac{1}{M}\sum_{m=1}^{M} H(P_t, P_s^{(m)}), \tag{4}$$

where $H$ denotes cross-entropy. This approach encourages the student to match the teacher's global representation of cardiac motion, despite observing only incomplete views of the video. Through video-level self-distillation, the student learns to recover the evolving dynamics of anatomical landmarks, capturing coherent motion patterns and structural features relevant to global heart function throughout the cardiac cycle.

## 3.2 Fine-Grained Online Spatial Guidance

Although video self-distillation promotes temporal consistency and global abstraction, it tends to overlook fine-grained spatial features, particularly those critical to clinical interpretation in echocardiography. Echocardiography imaging captures the dynamics and appearance of anatomically complex structures, where capturing subtle spatial details, such as mitral valve leaflet motion, septal wall thickness, or endocardial border definition, is crucial. To address this, we introduce a two-part strategy for enriching spatial detail and semantic structure in video representations:

**a). Masked Image Self-Distillation.** An online image encoder is trained to learn spatially rich features from partially masked images, enabling the extraction of fine-grained semantic concepts.
**b). Semantic Cluster Distillation (SCD).** A cross-modal clustering objective aligns reconstructed video tokens with spatial image features, encouraging the video model to organize its representation space around semantically meaningful structures.

### 3.2.1 Masked Image Self-Distillation

To learn fine-grained semantic features, we train an image encoder $\mathcal{I}_\theta$ in parallel with the video encoder. Each video $v$ is decomposed into individual frames $\{x_t\}$, which are processed independently. For each frame $x$, the teacher image encoder $\mathcal{I}_{\theta_t}$ receives the full-resolution image, while the student encoder $\mathcal{I}_{\theta_s}$ is given $N$ randomly masked variants $\{x_{\mathcal{M}_i}\}_{i=1}^N$. Each output is projected using distinct learnable heads $W_t$ (teacher) and $W_s$ (student), followed by softmax normalization:

$$P_s^{(i)} = \text{softmax}\left(\frac{W_s \mathcal{I}_{\theta_s}(x_{\mathcal{M}_i})}{\tau_s}\right), \quad P_t = \text{softmax}\left(\frac{W_t \mathcal{I}_{\theta_t}(x)}{\tau_t}\right), \tag{5}$$

where $\tau_s$ and $\tau_t$ are temperature parameters. The loss function encourages the student to match the teacher's predictions across all masked views:

$$\mathcal{L}_{\text{ssl}}^{img} = \frac{1}{N}\sum_{i=1}^N H(P_t, P_s^{(i)}), \tag{6}$$

with $H(\cdot, \cdot)$ denoting the cross-entropy. This training objective promotes the emergence of spatially grounded representations that encode fine-grained clinical concepts such as fetal heart valves, ventricular anatomy, and septal delineation that may be underrepresented in purely temporal learning.

### 3.2.2 Semantic Cluster Distillation (SCD)

While Masked Image Self-Distillation enables the image encoder to learn spatially grounded representations that capture fine-grained clinical concepts, it does not transfer this knowledge to the video encoder. As a result, the spatial and temporal representations remain disjoint. To bridge this gap, we introduce *Semantic Cluster Distillation (SCD)*, a cross-modal objective that distills semantic structure from the image encoder, guiding the video encoder to incorporate fine-grained spatial detail into its token representations.

Given a masked video input, the student video encoder $E_{\theta_s}$ processes the visible tokens to produce latent representations, which are then passed to a decoder $\psi$ that reconstructs token-level features $\hat{\mathbf{z}}_v \in \mathbb{R}^{B \times N \times D}$, where $B$ is the batch size, $N$ is the number of masked tokens, and $D$ is the feature dimension. In parallel, the corresponding video frames are processed by the image encoder $\mathcal{I}_{\theta_t}$, producing spatial features $\hat{\mathbf{z}}_i \in \mathbb{R}^{B \times N \times D}$. These image features are detached from the gradient flow and serve as semantic targets. Both sets of features are projected onto a shared set of learnable prototypes $P \in \mathbb{R}^{K \times D}$, resulting in similarity scores:

$$\mathbf{s}_v = \frac{\hat{\mathbf{z}}_v P^\top}{\tau}, \quad \mathbf{s}_i = \frac{\hat{\mathbf{z}}_i P^\top}{\tau}, \tag{7}$$

where $\tau$ is a temperature scaling parameter and $K$ is the number of prototypes. The resulting scores are transformed into Sinkhorn soft cluster targets using the Sinkhorn-Knopp algorithm:

$$\mathbf{q}_v = \text{Sinkhorn}(\mathbf{s}_v), \quad \mathbf{q}_i = \text{Sinkhorn}(\mathbf{s}_i). \tag{8}$$

The SCD loss symmetrically aligns the two modalities by minimizing the cross-entropy between their soft cluster assignments:

$$\mathcal{L}_{\text{SCD}} = \text{CE}(\mathbf{s}_v, \text{stopgrad}(\mathbf{q}_i)) + \text{CE}(\mathbf{s}_i, \text{stopgrad}(\mathbf{q}_v)), \tag{9}$$

Table 1: **Comparison of video anomaly detection methods on three echocardiography datasets**. Our method consistently outperforms SOTA approaches, demonstrating improved effectiveness in identifying cardiac abnormalities across diverse patient populations.

| Dataset | Model | Balanced Acc. | F1 | AUC |
|---------|-------|---------------|------|------|
| EchoNet-Dynamic | MNAD | 52.25 | 52.08 | 53.15 |
| | MemAE | 49.22 | 46.33 | 49.69 |
| | C2FPL | 57.36 | 57.35 | 59.00 |
| | Ours | **63.20** | **61.45** | **67.06** |
| RVENET | MNAD | 52.34 | 52.18 | 54.05 |
| | MemAE | 47.65 | 32.10 | 44.68 |
| | C2FPL | 47.88 | 47.86 | 46.30 |
| | Ours | **56.23** | **53.88** | **57.42** |
| Echo Pediatric-LVH | MNAD | 47.86 | 47.85 | 47.31 |
| | MemAE | 47.28 | 47.28 | 47.23 |
| | C2FPL | 51.39 | 51.31 | 50.68 |
| | Ours | **55.63** | **54.63** | **57.23** |

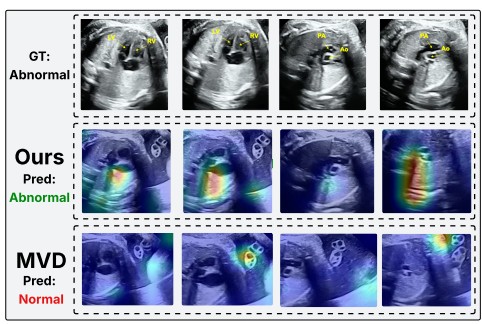

Figure 3: **Zero-Shot classification comparison:** (**Top**) The sweep from four-chamber to three-vessel view reveals smaller left-sided structures (LV and Ao) versus right-sided (RV and PA), consistent with coarctation of the aorta. (**Middle**) DISCOVR correctly identifies the abnormality, focusing on the ventricles in the four-chamber view and the Ao and PA in the vessel view. (**Bottom**)A backbone pretrained with MVD, in contrast, misclassifies the video as normal.

where gradients are propagated only through the video model and the prototype matrix $P$, while the image encoder is updated solely via its own self-distillation loss. This guides the video encoder to anchor its token representations to the spatially grounded clusters discovered by the image encoder, thereby distilling fine-grained anatomical detail into its temporal feature space. Semantic Cluster Distillation thus embeds spatial semantics within temporal features, yielding spatio-temporal representations that capture anatomically relevant detail in echocardiography videos.

## 4    Experiments and Results

**Datasets.** We use five ultrasound video datasets across fetal, pediatric, and adult populations. Two private fetal heart datasets, FetalEcho1 and FetalEcho2, were each collected from different hospital partners in the UK, comprising 10-second transverse, cephalad sweeps capturing five standard cardiac views (Situs, 4CH, LVOT, 3VV, 3VT). FetalEcho1 includes 8273/414/317 and FetalEcho2 includes 4154/320/305 videos for training/validation/testing. For adult and pediatric echocardiography, we use 3 public datasets: EchoNet Dynamic (apical 4CH adult; 7378/1326/1326) [28], EchoPediatric LVH (parasternal long-axis pediatric; 7837/1592/1592) [8], and RVENet (right ventricular pediatric/adult; 2516/487/573) [20]. Videos for adult and pediatric populations are labeled as *normal* or *abnormal* based on ejection fraction (EF), with *abnormal* defined as EF < 45% or EF > 75% [7]. Fetal videos are labeled as *normal* or *abnormal* based on expert evaluation by two fetal cardiologists(+10 years of experience). For the downstream segmentation task, we utilize the CAMUS [19] dataset.

**Evaluation.** All baseline models use official implementations, with videos sampled in 64-frame clips at a stride of 3. We adopt space-time tube embeddings from VideoMAE [35], treating each $2 \times 16 \times 16$ cube as a token with 90% masking ratio. All models use a ViT base backbone with consistent configurations. We evaluate representations using **zero-shot classification** and **linear probing**. Zero-shot evaluation uses a weighted kNN classifier [42, 4] on frozen features, with $k$ selected based on validation balanced accuracy. Linear probing trains a linear classifier for 30 epochs on a frozen backbone using a labeled validation set. During inference, each test video is divided into 64-frame clips and classified independently; a video is labeled abnormal if any clip is predicted abnormal. For segmentation evaluation, we add a linear layer followed by Conv2D upsampling blocks to generate pixel-level masks while keeping the backbone frozen.

**Baselines.** We compare DISCOVR with SOTA video SSL methods SIGMA [33], MGMAE [14], MVD [39], VideoMAE [35], and RAD-DINO [30], covering masked modeling, clustering, and dense feature learning. For anomaly detection, we include SOTA methods MNAD [29], MemAE [12],

Table 2: **Linear probing classification results on five echocardiography datasets** spanning fetal, adult, and pediatric populations. Our method achieves SOTA results, outperforming prior video SSL baselines and generalizing effectively across diverse clinical cohorts.

| Dataset | Model | Acc | Bal. Acc. | F1 |
|---|---|---|---|---|
| Fetal-Echo 1 | VideoMAE | 60.19 | 60.01 | 59.82 |
| | MGMAE | 59.55 | 59.40 | 59.30 |
| | SIGMA | 63.11 | 62.93 | 62.78 |
| | Ours | **65.70** | **65.52** | **65.39** |
| Fetal-Echo 2 | VideoMAE | 56.39 | 53.12 | 51.60 |
| | MGMAE | 60.98 | 60.49 | 60.43 |
| | SIGMA | 56.07 | 56.06 | 55.81 |
| | Ours | **65.25** | **63.53** | **63.59** |
| Echonet-Dynamic | VideoMAE | 71.04 | 70.86 | 70.85 |
| | MGMAE | 61.84 | 61.81 | 61.81 |
| | SIGMA | 75.57 | 75.48 | 75.50 |
| | Ours | **77.68** | **77.61** | **77.63** |
| Echo Pediatric-LVH | VideoMAE | 60.87 | 60.94 | 60.71 |
| | MGMAE | 54.71 | 51.70 | 49.46 |
| | SIGMA | 58.42 | 57.27 | 57.24 |
| | Ours | **62.81** | **61.64** | **61.66** |
| RVENET | VideoMAE | 60.03 | 60.31 | 59.70 |
| | MGMAE | 59.16 | 59.15 | 59.15 |
| | SIGMA | 59.51 | 59.25 | 58.98 |
| | Ours | **62.65** | **62.68** | **62.65** |

Table 3: **Zero-shot evaluation across five echocardiography datasets** covering fetal, adult, and pediatric populations. Our method consistently outperforms existing video SSL baselines, demonstrating robust generalization across diverse clinical populations.

| Dataset | Population | Model | Acc | Bal. Acc. | F1 |
|---|---|---|---|---|---|
| Fetal-Echo 1 | Fetal | RAD-DINO | 55.34 | 55.35 | 55.34 |
| | | VideoMAE | 60.52 | 60.81 | 60.00 |
| | | SIGMA | 54.37 | 54.91 | 51.90 |
| | | MGMAE | 60.84 | 61.03 | 60.64 |
| | | MVD | 59.87 | 60.20 | 59.15 |
| | | Ours | **62.46** | **62.79** | **61.79** |
| Fetal-Echo 2 | Fetal | RAD-DINO | 54.10 | 51.46 | 50.62 |
| | | VideoMAE | 50.49 | 48.01 | 47.21 |
| | | SIGMA | 55.41 | 51.90 | 49.92 |
| | | MGMAE | 59.34 | 56.71 | 56.09 |
| | | MVD | 59.34 | 55.45 | 53.14 |
| | | Ours | **59.67** | **57.18** | **56.69** |
| Echonet-Dynamic | Adult | RAD-DINO | 59.43 | 59.63 | 59.34 |
| | | VideoMAE | 57.16 | 57.91 | 55.07 |
| | | SIGMA | 53.47 | 54.46 | 49.04 |
| | | MGMAE | 51.21 | 52.23 | 46.13 |
| | | MVD | 60.11 | 60.94 | 57.56 |
| | | Ours | **62.59** | **63.20** | **61.45** |
| Echo Pediatric-LVH | Pediatric | RAD-DINO | 53.14 | 52.27 | 52.26 |
| | | VideoMAE | 51.57 | 53.98 | 50.47 |
| | | SIGMA | 47.55 | 49.56 | 46.80 |
| | | MGMAE | 46.61 | 48.91 | 45.45 |
| | | MVD | 49.56 | 51.91 | 48.46 |
| | | Ours | **54.65** | **55.63** | **54.63** |
| RVENET | Adult, Pediatric | RAD-DINO | 55.67 | 55.65 | **55.65** |
| | | VideoMAE | 54.97 | 55.64 | 52.24 |
| | | SIGMA | 52.36 | 53.18 | 47.64 |
| | | MGMAE | 53.23 | 54.08 | 48.17 |
| | | MVD | 54.62 | 55.12 | 53.17 |
| | | Ours | **55.67** | **56.23** | 53.88 |

and C2FPL [2], which rely solely on spatial-temporal learning without external modules like object detectors, pose estimators, or optical flow, often tailored to natural images.

## 4.1 Comparison with Video Anomaly Detection Methods

Table 1 compares the anomaly detection performance of DISCOVR with several state-of-the-art approaches. DISCOVR achieves the highest F1 score for all datasets (61.45% for EchoNet Dynamic, 53.88% for RVENET, and 54.63% for EchoPediatric LVH) as well as the highest balanced accuracy (63.20%, 56.23%, and 55.63%, respectively), substantially outperforming C2FPL, MemAE, and MNAD for all reported metrics. C2FPL relies on a multi-stage pseudo-labeling process to enhance anomaly discrimination, while both MemAE and MNAD incorporate sophisticated memory mechanisms and feature regularization in their inference pipelines. These methods employ targeted, anomaly-specific inference strategies and complex architectures.

In contrast, DISCOVR builds on a simple self-supervised learning framework that jointly learns spatial and temporal features, utilizing only a straightforward zero shot kNN classifier at inference. DISCOVR not only achieves state-of-the-art scores, including the highest AUCs of 67.06 on EchoNet Dynamic, 57.42 on RVENET, and 57.23 on EchoPediatric LVH, but also demonstrates that richer spatio-temporal representations learned via simple SSL can offer more effective and efficient anomaly detection than more sophisticaed anomaly detection techniques without reliance on specialized or resource intensive modules.

**Linear Probing.** Table 2 shows that DISCOVR achieves the highest balanced accuracy and F1 score in linear probing for anomaly detection across all echocardiography datasets. For example, on Echonet Dynamic, DISCOVR attains an F1 of 77.63 compared to 75.50 for SIGMA, and on FetalEcho 2, achieves 63.59 versus 60.43 for MGMAE. These improvements are consistent across fetal, pediatric, and adult cohorts. While VideoMAE and MGMAE rely on high masking ratios and pixel-level reconstruction, their representations often miss subtle anatomical landmarks and temporally distributed abnormalities, reflecting a lack of deeper semantic abstraction. Clustering-based approaches such as SIGMA can capture some temporal variation but lack explicit semantic guidance, limiting their ability to identify clinically relevant landmarks. In contrast, DISCOVR leverages semantic supervision from the image encoder through online distillation, combined with

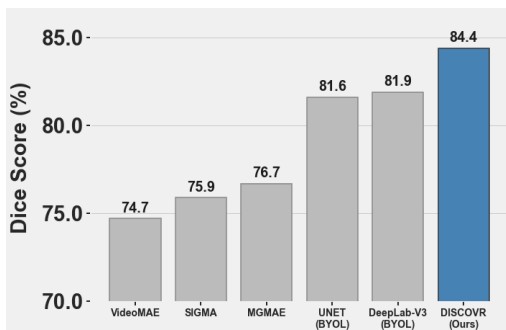

Figure 4: **Barplot comparing the segmentation performance across different models**. Our proposed DISCOVR approach achieves the highest Dice score of 0.844, outperforming both specialized segmentation architectures (DeepLab-V3, UNET) and other self-supervised methods.

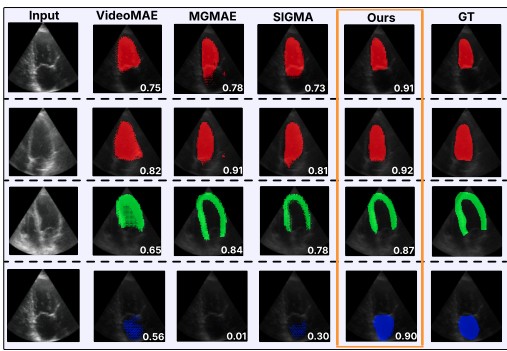

Figure 5: **Segmentation comparison on the CAMUS dataset** for left ventricular endocardium (LV Endo), left ventricular epicardium (LV Epi), and left atrium (LA). Our method produces accurate and consistent masks, achieving higher Dice scores (bottom right) than baseline methods.

temporal modeling in the video branch. This enables DISCOVR to capture fine-grained spatial features and their evolution over time, resulting in representations that are both robust and clinically meaningful for anomaly detection in cardiac ultrasound.

**Zero-Shot.** Table 3 shows that DISCOVR achieves the highest balanced accuracy and F1 score for zero shot classification across all echocardiography datasets. For example, on Echonet Dynamic, DISCOVR reaches an F1 of 61.45 compared to 57.56 for the best baseline, and on FetalEcho 1, achieves 61.79 versus 60.64 for MGMAE. These improvements are consistent across fetal, pediatric, and adult cardiac cohorts. This stronger performance reflects DISCOVR's ability to integrate semantic features captured by the image encoder with temporal dynamics modeled by the video branch, explicitly aligned through the SCD loss during self-supervised training. Pixel reconstruction models such as VideoMAE and MGMAE focus primarily on low-level appearance and texture, and clustering approaches like SIGMA, while using temporal clips, lack explicit semantic guidance. Image-based baselines like RAD-DINO do not leverage temporal information, while methods such as MVD that rely on external pretrained teachers may be less adaptable to the clinical and domain-specific challenges of ultrasound video. DISCOVR's capabilities are further highlighted in the qualitative example of Fig. 3, where it detects subtle cardiac structures and correctly classifies a challenging fetal video as abnormal, while MVD fails to capture these cues and predicts a normal outcome. This underscores how DISCOVR's features are sufficiently fine-grained to enable accurate zero shot anomaly detection, even without task-specific tuning.

## 4.2 Segmentation Evaluation

We evaluate the effectiveness of DISCOVR representations for downstream cardiac segmentation using the CAMUS dataset [19]. As shown in Fig 4, DISCOVR achieves the highest Dice score (0.844), outperforming specialized segmentation architectures such as UNet and DeepLabV3 (0.816 and 0.819, respectively, both with BYOL pretraining).When compared using a simple linear+upsampling head on a frozen backbone, DISCOVR also surpasses other SSL-based video models, including VideoMAE (0.747), MGMAE (0.767), and SIGMA (0.759). Fig. 5 highlights these advantages: DISCOVR produces consistently accurate and well-aligned segmentation masks for LV Endo, LV Epi, and especially the left atrium. For the challenging left atrium segmentation (blue mask), MGMAE misses the structure entirely (Dice = 0.01), while SIGMA and VideoMAE also perform poorly (Dice = 0.30 and 0.56). DISCOVR, in comparison, achieves 0.90, demonstrating superior ability to segment subtle structures and delineate boundaries due to its fine-grained feature learning.

## 4.3 LVEF Prediction

We evaluate the effectiveness of DISCOVR representations for downstream cardiac function estimation using the EchoNet-Dynamic ejection fraction dataset [28]. As shown in Table 4, DISCOVR achieves the lowest Mean Absolute Error (MAE) of 7.79 under the standard linear probing setup, outperforming other self-supervised baselines such as VideoMAE (8.02) and MGMAE (8.88). When

fine-tuning only the last three encoder blocks, DISCOVR further reduces the MAE to 6.32, demonstrating the strength of its learned representations even with limited adaptation. In comparison, the fully supervised EchoNet-Dynamic model [28] is trained end to end with all parameters updated. Under an ejection fraction–only setup without segmentation labels, DISCOVR surpasses these fully supervised baselines, including MC3 with an MAE of 6.59, the base EchoNet-Dynamic model with 7.35, and R3D with 7.63. The full EchoNet-Dynamic architecture achieves an MAE of 4.05 using a large multi-task design with 71.1 million parameters co-trained on 20,060 manual segmentation tracings. These results show that DISCOVR, through self-supervised pretraining and partial fine-tuning, learns powerful cardiac representations that rival or exceed fully supervised models trained end to end.

Table 4: LVEF prediction results on the EchoNet-Dynamic dataset. Our self-supervised method is compared against other SSL methods and fully-supervised baselines from [28].

| Model | MAE ↓ | RMSE ↓ | EF Labels | Seg. Labels |
|---|---|---|---|---|
| *Self-Supervised (Linear Probing)* | | | | |
| VideoMAE | 8.02 | 11.16 | ✓ | |
| MGMAE | 8.88 | 12.47 | ✓ | |
| DISCOVR (Ours) | **7.79** | **10.89** | ✓ | |
| *Self-Supervised (Fine-tuning)* | | | | |
| DISCOVR (finetune last 3 blocks) | **6.32** | **8.62** | ✓ | |
| *Fully-Supervised Baselines [1] trained only with EF Data* | | | | |
| MC3 (All frames) | 6.59 | 9.39 | ✓ | |
| EchoNet-Dynamic (EF, All frames) | 7.35 | 9.53 | ✓ | |
| R3D (All frames) | 7.63 | 9.75 | ✓ | |
| DISCOVR (finetune last 3 blocks,64 frames) | **6.32** | **8.62** | ✓ | |
| EchoNet-Dynamic (Full model) | **4.05** | **5.30** | ✓ | ✓ |

## 5 Ablation Study

In this section, we ablate the key components of the training objective in our model, **DISCOVR**. All experiments are conducted on the **Echonet Dynamic** dataset and evaluated using the **k-nearest neighbor (kNN) protocol**. This setup allows us to assess the discriminative quality of the learned representations in a fully frozen setting without additional fine-tuning.

**Effect of Loss Components.** We evaluate the effect of two core loss components used in DISCOVR: (i) the video self-distillation component ($\mathcal{L}_{ssl}^{vid}$), and (ii) the semantic cluster distillation component with online image guidance ($\mathcal{L}_{SCD}$). Table 5a reports the performance of these losses individually and in combination in zero-shot settings. Using only $\mathcal{L}_{ssl}^{vid}$ yields modest performance (F1 = 48.23%), as it primarily captures global temporal structure via CLS tokens but lacks guidance for fine-grained semantics. Introducing $\mathcal{L}_{SCD}$ leads to a substantial improvement (F1 = 61.45%, Balanced Accuracy = 63.20%), as the evolving image-based semantic clusters enrich the temporal features learned by the video model and encourage focus on more fine-grained, spatially grounded information. For more detailed ablation, refer to supplementary section B.1.4.

**Effect of Backbone Size.** We investigate how transformer backbone size impacts DISCOVR's representation quality. We evaluate ViT-Small and ViT-Base variants, each paired with matching DINO image encoders, on the Echonet Dynamic dataset using kNN evaluation (Table 5b). ViT-Base achieves superior performance (F1=61.45%, balanced accuracy=63.20%) compared to ViT-Small (F1=57.52%, balanced accuracy=59.44%). The smaller model's reasonable performance indicates DISCOVR learns meaningful representations even with limited capacity.

**Effect of Number of Frames.** In this ablation, we evaluate how the number of frames sampled from each video clip affects the representational quality learned by our model. We experiment with three temporal lengths: 16, 32, and 64 frames. All other training settings are kept constant, and the results are reported in Table 5d. We observe a clear upward trend in performance with increasing frame count. Using 16 frames results in an F1 score of 55.68%, which improves to 57.45% with 32 frames. The best performance is achieved with 64 frames, yielding an F1 score of 61.45% and balanced accuracy of 63.20%. These results support the intuition that ultrasound, being a temporally dense and

Table 5: Ablation studies reported with Balanced Accuracy, Precision, and F1 score.

(a) Effect of loss terms

| $\mathcal{L}_{\text{ssl}}^{vid}$ | $\mathcal{L}_{SCD}$ | Bal. Acc. | Precision | F1 |
|---|---|---|---|---|
| ✓ | ✗ | 52.27 | 53.16 | 48.23 |
| ✓ | ✓ | **63.20** | **65.35** | **61.45** |

(b) Backbone size

| Backbone | Bal. Acc. | Precision | F1 |
|---|---|---|---|
| ViT-Small | 59.44 | 61.03 | 57.52 |
| ViT-Base | **63.20** | **65.35** | **61.45** |

(c) Masking ratio

| Mask (%) | Bal. Acc. | Precision | F1 |
|---|---|---|---|
| 50 | 55.60 | 56.90 | 52.98 |
| 75 | 56.25 | 57.58 | 53.85 |
| 90 | **63.20** | **65.35** | **61.45** |

(d) Number of frames

| Frames | Bal. Acc. | Precision | F1 |
|---|---|---|---|
| 16 | 57.89 | 59.45 | 55.68 |
| 32 | 59.54 | 61.36 | 57.45 |
| 64 | **63.20** | **65.35** | **61.45** |

dynamic modality, benefits from longer clips. More frames provide richer temporal context, enabling the model to capture fine-grained spatial and temporal motion patterns across the cardiac cycle.

**Effect of Masking Ratio.** Table 5c shows a steady improvement in performance as the masking ratio increases, with F1-score rising from 52.98% (50%) to 61.45% (90%). Higher masking forces both the video encoder and the semantic image guidance branch to infer more from sparse visual cues, encouraging the model to focus on the most salient and non-redundant features. This promotes the learning of richer representations that better capture subtle and fine-grained spatio-temporal patterns, resulting in improved anomaly detection performance.

**Computational Cost and Scalability**. We report the computational cost, of our method in Table 6. The table shows both training and inference statistics, showing GPU memory usage and F1-score for each method on EchoNet-Dynamic, for a batch size of 1, 64 frames, and a spatial size of $112 \times 112$. During training, DISCOVR uses slightly more GPU memory (10.5GB) compared to prior methods (between 9.0 and 9.5GB) but achieves a notable +6.38% improvement in F1-score over the closest competitor. At inference, all methods, including DISCOVR, use identical ViViT-like encoders, resulting in nearly the same GPU memory footprint and FLOPS. This demonstrates that our method's performance improvements come with minimal extra training cost and no penalty for inference efficiency.

Table 6: Training and inference GPU memory, FLOPS, and F1-score on EchoNet-Dynamic, batch size = 1, 16 frames, $112 \times 112$ resolution.

| Model | Train Mem (GB) | F1-score | Infer Mem (GB) | Infer. FLOPS |
|---|---|---|---|---|
| MGMAE [14] | 9.0 | 46.13 | 1.153 | 101.85 |
| VideoMAE [35] | 9.0 | 55.07 | 1.153 | 101.85 |
| SIGMA [33] | 9.2 | 49.04 | 1.153 | 101.85 |
| Video-distillation | 9.5 | 48.23 | 1.153 | 101.85 |
| DISCOVR (Ours) | 10.5 | **61.45** | 1.153 | 101.85 |

# 6 Conclusion

We introduce DISCOVR, a self-supervised model for learning video representations in echocardiography across diverse patient populations. Our approach combines masked video modeling, temporal self-distillation, and online spatial supervision, unified by a Semantic Cluster Distillation (SCD) objective that aligns video and image features through cross-modal clustering, without relying on labeled anomalies or pretrained models. Extensively evaluated on six echocardiography datasets spanning fetal, pediatric, and adult populations, DISCOVR consistently outperforms previous self-supervised and anomaly detection methods for multiple tasks, including anomaly detection, classification (zero-shot and linear probing), and segmentation. DISCOVR's task-agnostic design and its applicability to diverse patient groups establish it as a strong foundation for screening cardiac conditions and developing assistive tools for echocardiography.

## Acknowledgments

We acknowledge financial support from InnoHK-funded Hong Kong Centre for Cerebro-cardiovascular Health Engineering (COCHE), UKRI grant EP/X040186/1, UK EPSRC grant EP/T028572/1 (VisualAI), UK EPSRC Doctoral Training Partnership award and, UKRI AIRR Early Access Project No. ANON-BYYG-VX4C-Z.

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

# Appendix

## A   Dataset Distribution

This section presents the dataset distributions for our five echocardiography video datasets: FetalEcho1 (Fig.6), FetalEcho2 (Fig.7), EchoNet-Dynamic (Fig.8), EchoNet-Pediatric (Fig.9), and RVENET (Fig.10). For each dataset, the bar chart displays the number of unique samples in the training, validation, and test sets. The accompanying pie charts illustrate the class distributions (Normal vs. Abnormal) within the validation and test sets.

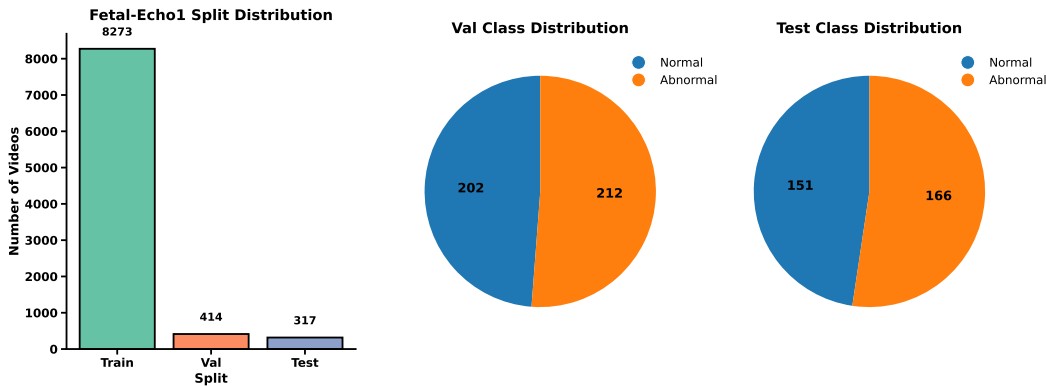

Figure 6: Dataset Distribution for Fetal-Echo1 dataset

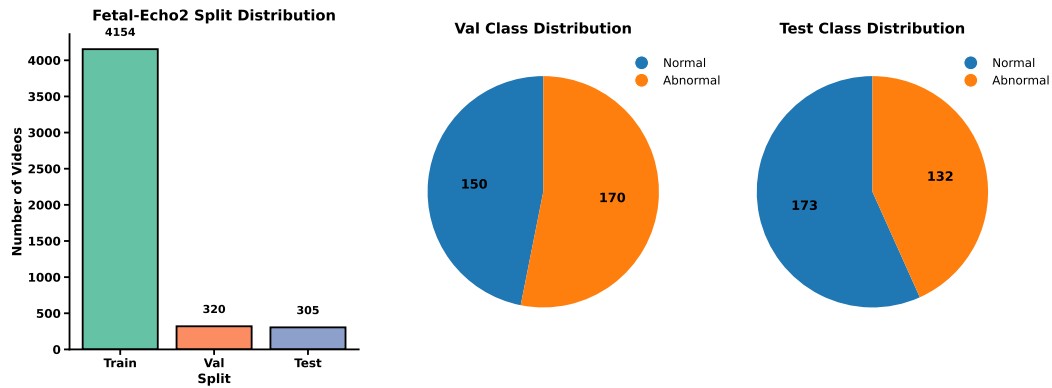

Figure 7: Dataset Distribution for Fetal-Echo2 dataset

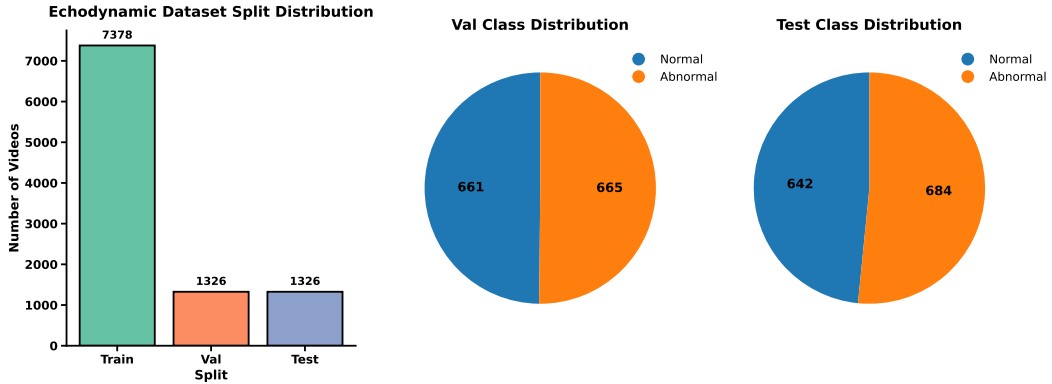

Figure 8: Dataset Distribution for Echo-Dynamic dataset

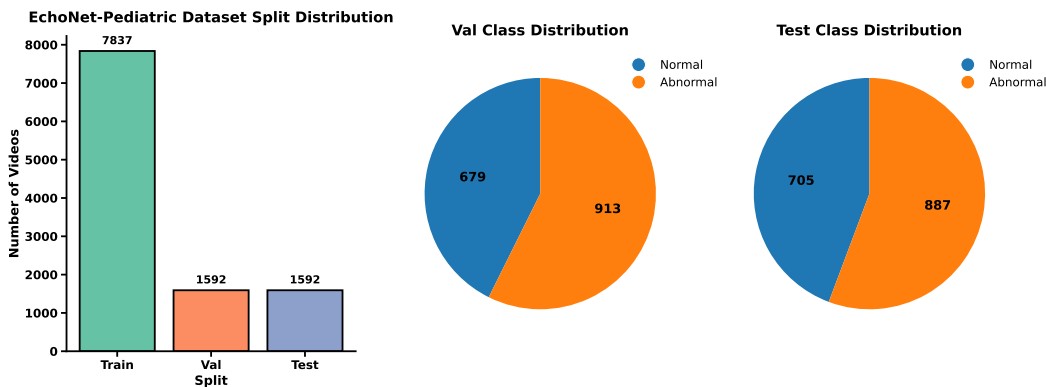

Figure 9: Dataset Distribution for Echo-Pediatric dataset

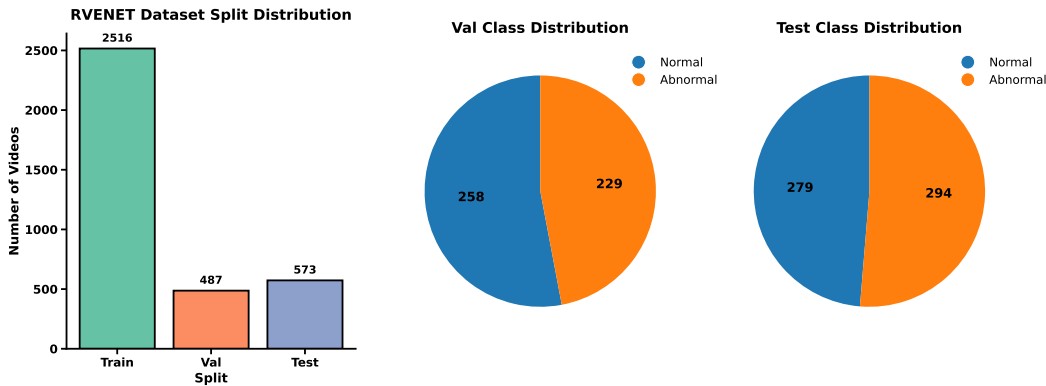

Figure 10: Dataset distribution for RVENET dataset

# B  Additional Results:

## B.1  Full Finetuning

### B.1.1  Evaluation Setup

We follow the same evaluation procedure as described in the experiments section in the main paper, but fine-tune the entire backbone along with the linear layer. All other evaluation settings remain

unchanged. Results are reported on the Echonet Dynamic dataset to assess end-to-end supervised performance.

### B.1.2 Evaluation Result:

Under full fine-tuning, as shown in Table 7, all models experience a drop in performance compared to their linear probing results, reflecting overfitting due to the limited labeled validation data. Despite this, DISCOVR achieves the highest F1 score of 70.44%, outperforming MGMAE (65.99%), SIGMA (61.46%), and VideoMAE (57.31%). DISCOVR's structured representation learning, through temporal distillation and cross-modal clustering, appears to provide more robust and adaptable features, enabling it to generalize better even when fully fine-tuned on a small dataset.

Table 7: Table showing the full-finetuning result of DISCOVR compared to other baselines on the Echo-Dynamic Dataset

| Model (%) | Accuracy | Balanced Acc. | Precision | Recall | F1-Score |
|---|---|---|---|---|---|
| VideoMAE | 57.62 | 57.94 | 58.27 | 57.94 | 57.31 |
| SIGMA | 61.69 | 62.00 | 62.41 | 62.00 | 61.46 |
| MGMAE | 65.99 | 66.08 | 66.10 | 66.08 | 65.99 |
| Ours | **70.51** | **70.42** | **70.50** | **70.42** | **70.44** |

### B.1.3 Generalisation to Other Modalities.

To test whether DISCOVR generalizes beyond echocardiography, we evaluated its transfer performance on two distinct medical image benchmarks: the Breast Ultrasound Images dataset [1] (cancer detection across 600 patients) and DermMNIST [44] (skin lesion classification). Both breast ultrasound and echocardiography require the detection of small, irregular regions of altered tissue, such as hypoechoic tumors in the breast or localized wall motion abnormalities in the heart, making the ability to identify subtle structural changes in one domain directly applicable to the other. Similarly, DermMNIST demands fine-grained visual discrimination between morphologically similar skin lesions. For both benchmarks, we froze the DISCOVR encoder and trained a linear classifier, comparing performance directly across methods.

As shown in Table 8, our method demonstrates strong generalization across both tasks. On the Breast Ultrasound dataset, DISCOVR improves balanced accuracy by 2.01% over VideoMAE, 19.83% over SIGMA, and 12.01% over MGMAE. For DermMNIST, DISCOVR achieves an accuracy of 71.68%, outperforming VideoMAE by 2.85%, SIGMA by 3.00%, and MGMAE by 3.85%. These results demonstrate strong generalization to multiple medical image analysis tasks beyond echocardiography. Further, to assess generalization to natural video data, we pretrained and evaluated all models on the Kinetics 400 action recognition benchmark, using a zero-shot protocol where KNN classification with $K = 20$ was applied to features using 64 frames from the frozen video backbone. As shown in Table 9, DISCOVR achieves the highest Top-1 accuracy at 22.3%, outperforming MVD by 3.6%, MME by 3.2%, and VideoMAE by 1.6%, while also requiring the fewest pretraining epochs. These results highlight that DISCOVR not only excels at medical video tasks but also learns generalizable representations efficiently for large-scale natural video datasets.

Table 8: Linear Probing results on the **Breast Ultrasound** dataset and **DermMNIST**. For Breast Ultrasound, we report Balanced Accuracy (Bal. Acc.) and F1; for DermMNIST, we report overall Accuracy (Acc.).

| Method | Breast Ultrasound | | DermMNIST |
|---|---|---|---|
| | Balanced Accuracy | F1 Score | Accuracy |
| VideoMAE | 61.75 | 64.45 | 68.83 |
| SIGMA | 43.93 | 42.21 | 68.68 |
| MGMAE | 51.75 | 52.34 | 67.83 |
| **DISCOVR (Ours)** | **63.76** | **65.44** | **71.68** |

Table 9: Zero-Shot KNN classification performance on Kinetics-400

| Model | Epochs | Top-1 Accuracy (%) |
|---|---|---|
| MVD [39] | 1600 | 18.7 |
| MME [34] | 800 | 19.1 |
| VideoMAE [35] | 800 | 20.7 |
| **DISCOVR (Ours)** | **400** | **22.30** |

#### B.1.4  Loss function Ablation detailed.

To rigorously evaluate each component, we have added baselines using only masked image or only video self-distillation. Indeed, we find that the settings perform suboptimally, as shown in Table 1, confirming that spatial or temporal cues alone are insufficient for strong representation learning. In contrast, combining both with the SCD loss, which explicitly distills fine-grained semantic structure from the image branch into the video backbone, achieves the best results. This supports our intuition that SCD is crucial for aligning spatial semantics with temporal dynamics, enabling more robust and clinically meaningful video representations.

Table 10: Effect of the different loss terms on classification performance (Balanced Accuracy, Precision, and F1).

| $\mathcal{L}_{\text{ssl}}^{vid}$ | $\mathcal{L}_{\text{ssl}}^{img}$ | $\mathcal{L}_{\text{SCD}}$ | Bal. Acc. | Precision | F1 |
|---|---|---|---|---|---|
| ✓ | ✗ | ✗ | 52.27 | 53.16 | 48.23 |
| ✗ | ✓ | ✗ | 53.66 | 55.22 | 49.43 |
| ✓ | ✓ | ✓ | **63.20** | **65.35** | **61.45** |

## C  Implementation Details

All models are implemented in PyTorch 2.6 and trained on RTX 8000 GPUs (48 GB) with a batch size of 8 using the AdamW optimizer. Videos are processed as 64-frame clips sampled at a stride of 3 and resized to $112 \times 112$.

For both video and image self-distillation, we use a student-teacher setup where the teacher processes the full input and the student observes $N = 4$ randomly masked views. The teacher network is updated via an exponential moving average (EMA) of the student with momentum $\lambda = 0.996$. A fixed temperature $\tau_s = 0.1$ is used for the student, while the teacher temperature $\tau_t$ is linearly warmed from 0.04 to 0.07 over the first 30 epochs. Semantic Cluster Distillation (SCD) uses $K = 3000$ learnable prototypes, with similarity scores computed via temperature-scaled dot products ($\tau = 0.1$) and cluster assignments generated using the Sinkhorn-Knopp algorithm (10 iterations, $\epsilon = 0.05$). Models are trained for 400 epochs with a learning rate of $1.5 \times 10^{-4}$, weight decay of 0.05, and 40 warmup epochs.

## D  Broader Impact and Limitations

In this work, we introduce **DISCOVR**, a novel self-supervised model for echocardiography video understanding across fetal, pediatric, and adult populations. Trained without labeled abnormal cases, DISCOVR learns rich spatiotemporal representations and enables zero-shot inference. One key application of DISCOVR is in the *early screening of heart diseases*, where it can assist clinicians by flagging potential anomalies in echocardiography videos. This has significant clinical relevance, as congenital heart defects affect approximately 1 in 100 newborns, with up to 50% missed during prenatal screening [36, 17, 37], and cardiovascular diseases remain the leading global cause of death [40]. By reducing reliance on large, labeled datasets, DISCOVR offers a scalable and accessible solution, particularly for deployment in low-resource settings.

While DISCOVR shows strong potential, its current scope is focused specifically on echocardiography, and it has not yet been evaluated on other imaging modalities. The model was trained and tested on

five datasets collected from distinct clinical sites, each with its own imaging protocols, devices, and patient cohorts. As a result, the demographic and geographic diversity of the data may be limited. Further validation is needed to assess the model's generalizability across broader clinical settings, populations, and imaging systems.

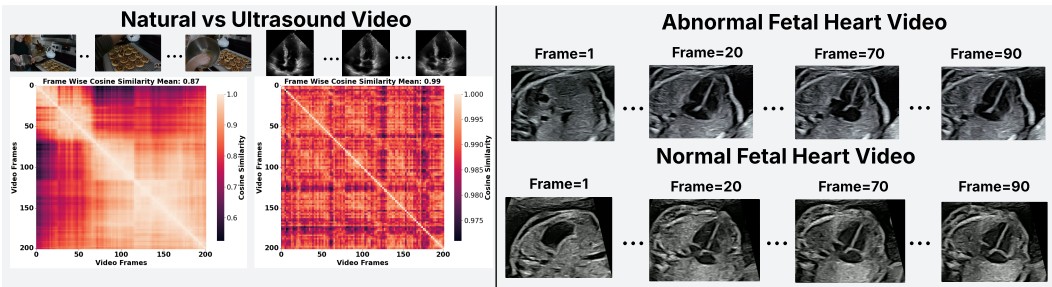

Figure 11: Figure (left) compares two fine-grained videos: a natural scene of a person baking (left) and an adult heart ultrasound (right). The frame-level cosine similarity matrix, computed using a pretrained VideoMAE model, shows that ultrasound frames are highly similar (mean=0.99), with only minor local variations. This highlights the difficulty in distinguishing individual frames in such medical videos. Figure (right) compares normal and abnormal fetal echocardiograms, which appear almost identical despite one being abnormal. This illustrates the inherent difficulty of distinguishing subtle cardiac abnormalities in fetal imaging.

