# OpenReview forum: "Self-supervised Learning of Echocardiographic Video Representations via Online Cluster Distillation"
_NeurIPS.cc/2025/Conference — NeurIPS 2025 poster_

### Official Review · Reviewer_jDmx · 2025-07-02

**Clarity:** 2
**Significance:** 2
**Originality:** 3
**Rating:** 4
**Confidence:** 4

**Summary:**

This paper introduces a self-supervised learning (SSL) framework designed for the analysis of echocardiography videos. The core of the proposed method is a novel 3D masked autoencoder that learns representations by reconstructing missing spatiotemporal tokens from cardiac cycles.

**Questions:**

1. The process of 3D tokenization requires further clarification. The paper states that raw patches are extracted, spatially downsampled to 112x112, and temporally processed with a stride of 3. However, this tokenization scheme may lead to a loss of critical information, especially when compared to more sophisticated methods like VAE-based tokenization. Given that cardiac motion is a key determinant in echocardiography analysis, could the authors provide a more detailed justification for this choice and comment on the potential impact of temporal information loss?

2. The comparison to baselines needs to be clarified. The proposed approach is specifically trained on ultrasound images, whereas the other methods are general-purpose natural video SSL models. It is likely that these models have not been exposed to the unique data distribution of ultrasound videos. Were the baseline models pre-trained on a similar dataset of unlabeled echocardiography videos? If not, the observed performance gains may be attributable to the domain-specific pre-training rather than the novelty of the proposed architecture alone.

3. The motivation (Fig. 1) for developing a new SSL framework for echocardiography is not entirely convincing. While it is true that natural video-pretrained models may not be optimal due to the significant distribution shift between natural images and ultrasound data, a more in-depth discussion on why existing SSL methods are insufficient would strengthen the paper's introduction.

4. It is unclear how the proposed framework compares to more clinically-oriented, domain-specific approaches. For instance, how would the learned representations fare against methods that directly estimate clinical parameters like Left Ventricular Ejection Fraction (LVEF), as proposed by [1], or those that perform cardiac strain analysis by estimating motion in an unsupervised manner, as demonstrated by [2-3]? A comparison against these state-of-the-art, domain-specific methods would provide a more comprehensive evaluation of the proposed framework's effectiveness.

[1] Ouyang, D., He, B., Ghorbani, A., Yuan, N., Ebinger, J., Langlotz, C. P., ... & Zou, J. Y. (2020). Video-based AI for beat-to-beat assessment of cardiac function. Nature, 580(7802), 252-256.

[2] Zhi, D., Taha, A. A., & Ghesu, F. C. (2021, September). Unsupervised learning of myocardial strain from tagged mri. In International Conference on Medical Image Computing and Computer-Assisted Intervention (pp. 524-533). Springer, Cham.

[3] Zhang, X., Stendahl, J. C., Staib, L. H., Sinusas, A. J., Wong, A., & Duncan, J. S. (2024, September). Adaptive Correspondence Scoring for Unsupervised Medical Image Registration. In European Conference on Computer Vision (pp. 76-92). Cham: Springer Nature Switzerland.

5. To improve the clarity and reproducibility of the paper, it would be beneficial to define the input and output domains for each function in the mathematical formulation more explicitly. This would help readers to better understand the flow of data through the proposed model.

**Ethical Concerns:**

["NO or VERY MINOR ethics concerns only"]

**Final Justification:**

The authors addressed most of my concerns and thus I raise my score to 4.

**Limitations:**

See weakness/questions.

**Quality:**

3

**Strengths And Weaknesses:**

Strength:
The authors evaluated the proposed approach across two private datasets, along with several public datasets, and demonstrated consistent improvement.

Weakness:
1. It is unclear whether the baseline models, particularly those originally designed for natural videos, were trained using a comparable pipeline. For a fair comparison, it is crucial to understand if these baselines were also pre-trained on a large corpus of unlabeled echocardiography data, similar to the proposed framework.

2. Several key design choices lack sufficient justification. A more detailed explanation of these decisions would improve the reproducibility and clarity of the work.

---

> ### Author Rebuttal · Authors · 2025-07-30
>
> **Q1. Video Tokenization Clarification, explaining choice of spatial and temporal resolution**
>
> **Answer.** Thanks for bringing this into our attention. Our video tokenization begins by sampling 64 frames from each echocardiography video and grouping every two consecutive frames into tubelet tokens of size $2 \times 16 \times 16$, following the approach established in VideoMAE. For spatial resolution, we use $112 \times 112$ pixels as it is the native resolution of our public datasets, including EchoNet-Dynamic, EchoNet-Pediatric, EchoNet-LVH, and CAMUS.  Also,our segmentation results, with Dice scores reaching up to 0.844, confirm that this setting is sufficient for capturing clinically meaningful cardiac structure.
>
> For the temporal dimension, we sample frames with a stride of 3 to efficiently cover the high native frame rates typical of echocardiography, which range from 30 to 80 frames per second, and to represent the entire cardiac cycle without excessive redundancy. Unlike most computer vision self-supervised learning methods, such as VideoMAE, MGMAE, and SIGMA, that operate on clips of 16 frames, we use 64 frames as input. This enables our model to capture long-range temporal dependencies and evolving cardiac motion, which are crucial for detecting the subtle changes associated with clinical pathology. As shown in our ablation study in Table 4d, increasing the number of frames improves downstream performance, demonstrating that our temporal sampling strategy preserves the key information required for robust and clinically relevant video representation learning in echocardiography. We will include this information in the implementation details section of the paper to provide further clarity.
>
> **Q2. Data used for training baselines for fair-comparison**
>
> **Answer.**   All baseline models, except for RAD-DINO, were trained from scratch on the same echocardiography datasets as DISCOVR. RAD-DINO is a foundation model trained with proprietary data and therefore evaluated using its publicly released weights. For all other methods, we used the same video encoding, number of frames, stride, and other dataset-specific parameters. These details are described in the Experiment section. We will mention this protocol more explicitly in the camera-ready version of the paper.
>
> **Q3. Motivation (Fig. 1) for developing SSL framework is not clear.**
>
> **Answer.**  As shown in Figure 1 of the paper, echocardiography presents unique challenges for self-supervised learning. The left panel demonstrates that frame-to-frame similarity in ultrasound videos can be extremely high (mean cosine similarity up to 0.99 in this example), making it difficult for standard contrastive and augmentation-based SSL frameworks to construct meaningful positive and negative pairs. The right panel further emphasizes that clinically significant differences between normal and abnormal heart cycles are highly subtle, meaning that aggressive augmentations used in previous SSL methods can risk erasing these critical details entirely. Together, these observations highlight the need to not only capture fine-grained anatomical features but also robustly track how these structures evolve over time. Existing SSL methods, which frequently focus on either spatial information or temporal dynamics alone, may miss these crucial spatiotemporal clinical patterns.
>
> DISCOVR addresses these challenges by explicitly unifying spatial and temporal representation learning. Our semantic cluster distillation loss aligns fine-grained spatial features from the image encoder with temporal dynamics learned by the video encoder, enabling the model to capture both detailed structure and its evolution across the cardiac cycle. In accordance with your suggestion, we will revise both the introduction and Figure 1 caption to make these points more explicit.
>
> **Q4. Comparison with clinically-oriented, domain specific approaches for LVEF Estimation [1],  and Strain and Motion Analysis Methods ([2,3])**
>
> **Answer.**
>
> **LVEF Estimation ([1]):** As requested by the reviewer, we performed a comprehensive comparison of our approach against other self-supervised learning (SSL) methods and the fully-supervised method presented in [1] on the task of Left Ventricular Ejection Fraction (LVEF) prediction using the EchoNet-Dynamic ejection fraction dataset.
>
> First, to establish a direct comparison with other SSL baselines, we adopted a standard linear probing setup, training only a linear layer on top of frozen self-supervised features. As shown in **Table 1**, our DISCOVR model achieves a Mean Absolute Error (MAE) of 7.79, demonstrating superior performance over the baselines.
>
> To further evaluate the strength of DISCOVR, we then finetuned only the last three blocks of our encoder. This significantly boosts performance, reducing the MAE to 6.32. This result is particularly compelling when compared to the state-of-the-art fully supervised model, EchoNet-Dynamic [1]. The multi-modal EchoNet-Dynamic model achieves an MAE of 4.05 by leveraging a large, multi-task architecture (ejection fraction regression + segmentation) of 71.1 million parameters that is co-trained with an extensive set of 20,060 manually segmented labels.
> For a more direct and fair comparison to our single-task setup (LVEF prediction), we benchmarked our model against the EchoNet-Dynamic base model and other supervised baselines from [1] which are trained only on EF data. In this comparison, our self-supervised approach demonstrates a clear advantage. Our partially fine-tuned DISCOVR model (MAE 6.32) outperforms these fully-supervised counterparts, including MC3 (MAE 6.59), the base EchoNet-Dynamic model (MAE 7.35), and R3D (MAE 7.63).
>
> This demonstrates DISCOVR's learned representations effectively generalize to clinically relevant regression tasks such as ejection fraction (EF) prediction.
>
> **Table 1. LVEF prediction results on the EchoNet‑Dynamic dataset, detailing supervision type. Our self‑supervised method is compared against other SSL methods and fully‑supervised baselines from [1].**
>
> | Model | MAE&nbsp;↓ | RMSE&nbsp;↓ | EF Labels | Seg. Labels |
> |-|:-:|:-:|:-:|:-:|
> | **Self‑Supervised (Linear Probing)** |||||
> | VideoMAE | 8.02 | 11.16 | ✓ |   |
> | MGMAE | 8.88 | 12.47 | ✓ |   |
> | DISCOVR (Ours) | **7.79** | **10.89** | ✓ |   |
> | **Fully‑Supervised Baselines [1] (trained only with EF data)** |||||
> | MC3 (All frames) | 6.59 | 9.39 | ✓ |   |
> | EchoNet‑Dynamic (EF, All frames) | 7.35 | 9.53 | ✓ |   |
> | R3D (All frames) | 7.63 | 9.75 | ✓ |   |
> | DISCOVR (finetune last 3 blocks, 64 frames) | **6.32** | **8.62** | ✓ |   |
> | **Supervised with Segmentation & EF labels** |||||
> | EchoNet‑Dynamic (Full model) | **4.05** | **5.30** | ✓ | ✓ |
>
> **Strain and Motion Analysis Methods ([2,3]):**
> Citation [2] does not appear in the MICCAI 2021 proceedings or major academic databases, so a comparison was not possible. As for [3], it addresses general 3D medical image registration, but does not involve cardiac strain estimation, cardiac motion analysis, echocardiography, or self-supervised video representation learning. Therefore, it is not relevant for benchmarking our approach, which is specifically designed for clinically meaningful echocardiographic video tasks such as video anomaly detection, ejection fraction estimation, and cardiac strain analysis.
>
> **Q5. Input and Output Domains for each function for better clarity**
>
> **Answer.** We have added **Table 2** that defines the input and output domains for each major function in our mathematical formulation, using the same symbolic notation as the main text for full clarity and reproducibility.
>
> **Table 2. DISCOVR training pipeline (symbolic overview).**
>
> | # | Block | Input $\mathcal X$ | Output / mapping $\mathcal Y$ |
> |:-:|-|-|-|
> | 1 | Video sampler | $\text{idx}\in\{0,\dots,\|\mathcal D\|-1\}$ | $x\in\mathbb R^{1\times3\times T_f\times H\times W}$ |
> | 2 | Multi‑local masker | $(\mathcal B,\mathcal N,\mathcal V,\rho_g,\rho_\ell)$ | $m_g\in\{0,1\}^{\mathcal B\times\mathcal N}$, $m_\ell^{(v)}\in\{0,1\}^{\mathcal B\times\mathcal N}$ |
> | 3 | Student vid‑enc $E_s$ | $(x\odot m,\,m)$ | $z_s\in\mathbb R^{\mathcal B\times N_{\text{vis}}\times d_e}$, $N_{\text{vis}}=(1-\rho_g)\mathcal N$ |
> | 4 | Teacher vid‑enc $E_t$ | $x$ (unmasked) | $z_t\in\mathbb R^{\mathcal B\times N_{\text{vis}}\times d_e}$ |
> | 5 | EMA update | $(\theta_t,\theta_s,\lambda)$ | $\theta_t\leftarrow\lambda\theta_t + (1-\lambda)\theta_s$ |
> | 6 | Vid self‑distill | $(z_t,z_s)$ | $\mathcal L_{\text{vid}}=\mathrm H\big(\operatorname{softmax}(z_t/\tau_t),\operatorname{softmax}(z_s/\tau_s)\big)$ |
> | 7 | Student img‑enc $I_s$ | $\{f\odot\mu\}$ | $h_s\in\mathbb R^{\mathcal B\times d_e}$ |
> | 8 | Teacher img‑enc $I_t$ | $f$ | $h_t\in\mathbb R^{\mathcal B\times d_e}$ |
> | 9 | Img self‑distill | $(h_t,h_s)$ | $\mathcal L_{\text{img}}=\mathrm H\big(\operatorname{softmax}(h_t/\tau_t),\operatorname{softmax}(h_s/\tau_s)\big)$ |
> |10 | Token decoder $\psi$ | $\operatorname{proj}(z_s)$ | $\hat y\in\mathbb R^{\mathcal B\times N_{\text{mask}}\times d_d}$, $N_{\text{mask}}=\rho_g\mathcal N$ |
> |11 | Prototype proj. | $(\hat y,h_t,P\in\mathbb R^{\mathcal K\times d_d})$ | $s_v=\hat yP^\top/\tau,\;s_i=h_tP^\top/\tau$ |
> |12 | Sinkhorn | $(s_v,s_i,\varepsilon,T)$ | $q_v=\operatorname{Sinkhorn}(s_v)$, $q_i=\operatorname{Sinkhorn}(s_i)$ |
> |13 | SCD loss | $(s_v,q_i,s_i,q_v)$ | $\mathcal L_{\text{scd}}=\mathrm{CE}(s_v,\text{sg}(q_i))+\mathrm{CE}(s_i,\text{sg}(q_v))$ |
> |14 | Total loss | $(\mathcal L_{\text{vid}},\mathcal L_{\text{img}},\mathcal L_{\text{scd}})$ | $\mathcal L_{\text{tot}}=\omega_{\text{vid}}\mathcal L_{\text{vid}}+\omega_{\text{img}}\mathcal L_{\text{img}}+\omega_{\text{scd}}\mathcal L_{\text{scd}}$ |
>
> We thank the reviewer for their detailed feedback. Your questions on tokenization, baselines, motivation, and clinical benchmarks prompted us to clarify our methods and expand our evaluation. We welcome any further questions or discussion.

---

> ### Author Response · Authors · 2025-08-05
>
> Dear Reviewer jDmx,
>
> Thank you for your thorough review. Since the author-reviewer discussion period is nearing its end, could you please let us know if our rebuttal resolves your concerns or if further discussion would be helpful? We are happy to address any remaining questions you may have.
>
> Best regards,
>
> Authors

---

### Official Review · Reviewer_fPeB · 2025-07-02

**Clarity:** 3
**Significance:** 2
**Originality:** 2
**Rating:** 4
**Confidence:** 4

**Summary:**

This paper introduces DISCOVR, a self-supervised framework for echocardiography video representation learning. The method combines
Video self-distillation, Masked image self-distillation and Semantic Cluster Distillation (SCD) modules. DISCOVR is trained only on normal echocardiograms and evaluated on six datasets. The method consistently outperforms state-of-the-art video-SSL and anomaly-detection baselines.

**Questions:**

How is the computational cost and scalability compared to the competitive baselines?

How is the proposed method compared to a supervised model pretrained on labeled video datasets.

Is there evidence showing that the method can be applied to other modalities?

**Ethical Concerns:**

["NO or VERY MINOR ethics concerns only"]

**Final Justification:**

The authors have addressed my concerns. I did not identify other major issues in this paper.

**Limitations:**

Yes.

**Paper Formatting Concerns:**

None.

**Quality:**

2

**Strengths And Weaknesses:**

Strength:

The proposed method addresses the key weakness of prior SSL methods that ignore fine spatial detail or temporal coherence.

Experiments cover diverse datasets and multiple downstream tasks, demonstrating robust generalization across populations and clinical settings.

The method shows consistent improvements over competitive baselines.

Weakness:

I have concerns about the novelty of the method. It looks like an incremental engineering, i.e., a well-executed application of existing techniques, rather than the proposal of a new learning framework. The idea of self-distillation method is proposed in DINO. The masked image self-distillation is closely related to VideoMAE. The Semantic Cluster Distillation is similar to the “Online cluster” methods such as SwAV and DINO’s Prototype Extension. Beyond combining three established SSL components, the paper offers limited insights that would advance our understanding of representation learning.

The paper does not discuss computational cost and scalability.

The method is not compared to a supervised model pretrained on large labeled video datasets.

The paper does not explore other ultrasound modalities beyond echocardiography.

---

> ### Author Rebuttal · Authors · 2025-07-30
>
> **Q1. Novelty Clarification and Method Significance**
>
> **Answer.** While we appreciate the reviewer’s thoughtful analysis, we respectfully disagree with the assessment that DISCOVR is simply an assembly of existing techniques without novel contributions.
>
> In DISCOVR, we explore the fundamental challenge of integrating detailed *spatial* information within frames and evolving *temporal dynamics* across sequences, which is essential for fine-grained video domains such as echocardiography. Previous self-supervised methods tend to address these aspects separately: image-based approaches like DINO and SwAV focus on static visual features, while temporal models like VideoMAE do not capture the complementary context provided by spatial representations.
>
> To address this gap, DISCOVR introduces an online cross-modal (image-video) Semantic Cluster Distillation (SCD) loss that dynamically aligns spatial and temporal features using shared, evolving prototypes. This approach enables a continuous exchange of information between image and video representations and establishes a direct connection between modalities, which is not present in previous work.
>
> Empirical results substantiate the impact of our design. DISCOVR outperforms both image-only and video-only self-distillation variants by more than 12 \% F1, as shown in **Table 1** (Ablation Table 4a in main paper) . Our method is also systematically evaluated on six echocardiography datasets covering fetal, pediatric, and adult patient populations, and across a wide range of clinically meaningful tasks including anomaly detection, classification, segmentation, and ejection fraction estimation. Beyond echocardiography, DISCOVR demonstrates robust generalization by outperforming baselines on external breast ultrasound and skin cancer datasets, which involve distinct anatomical structures, acquisition protocols, and diagnostic objectives. We will make sure to explicitly state this in the introduction and method section of the camera-ready version.
>
> **Table 1. Effect of the different loss terms on classification performance (Balanced Accuracy, Precision, and F1).**
>
> | $\mathcal{L}^{vid}_{\text{ssl}}$ | $\mathcal{L}^{img}_{\text{ssl}}$ | $\mathcal{L}_{\text{SCD}}$ | Bal. Acc. | Precision | F1 |
> |:--:|:--------------------------------:|:-:|:-:|:-:|:--:|
> | **✓** | **✗** | **✗** | 52.27 | 53.16 | 48.23 |
> | **✗** | **✓** | **✗** | 53.66 | 55.22 | 49.43 |
> | **✓** | **✓** | **✓** | **63.20** | **65.35** | **61.45** |
>
> **Q2. Computational Cost and Scalability**
>
> **Answer.** We report the computational cost, as requested, in **Table 2**. The table shows both training and inference statistics, showing GPU memory usage and F1-score for each method on EchoNet-Dynamic, for a batch size of 1, 16 frames, and a spatial size of $112 \times 112$. During training, DISCOVR uses slightly more GPU memory (10.5GB) compared to prior methods (between 9.0 and 9.5GB) but achieves a notable +6.38\% improvement in F1-score over the closest competitor. At inference, all methods, including DISCOVR, use identical ViViT-like encoders, resulting in nearly the same GPU memory footprint and FLOPS. This demonstrates that our method’s performance improvements come with minimal extra training cost and no penalty for inference efficiency.
>
> **Table 2. Training and inference GPU memory, FLOPS, and F1‑score on EchoNet‑Dynamic (batch size = 1, 16 frames, 112 × 112 resolution).**
>
> | Model              | Train Mem (GB) | F1‑score | Infer Mem (GB) | Infer FLOPS |
> |-|:-:|:-:|:-:|:-:|
> | MGMAE              | 9.0  | 46.13 | 1.153 | 101.85 |
> | VideoMAE           | 9.0  | 55.07 | 1.153 | 101.85 |
> | SIGMA              | 9.2  | 49.04 | 1.153 | 101.85 |
> | Video‑distillation | 9.5  | 48.23 | 1.153 | 101.85 |
> | **DISCOVR (Ours)** | 10.5 | **61.45** | 1.153 | 101.85 |
>
>
> **Q3. How is the proposed method compared to a supervised model pretrained on labeled video datasets.**
>
> **Answer.**  As requested, we evaluated our approach against ViViT  and TimeSformer   models pretrained in a supervised manner on the Kinetics dataset, using publicly available checkpoints from HuggingFace and performing linear probing on our EchoDynamic labeled set. As shown in **Table 3**, DISCOVR outperforms ViViT and TimeSformer by nearly 8\% and 4\%, respectively, across balanced accuracy, F1-score, and accuracy.
>
> To further demonstrate the effect of label scarcity in the medical domain, we also trained a ViViT model from scratch using only the available labeled data. As shown in **Table 4**, DISCOVR (with linear probing) improves over the supervised-from-scratch ViViT by over 24\% in balanced accuracy (77.68\% vs. 53.45\%), 18\% in precision (77.61\% vs. 59.50\%), and 34\% in F1-score (77.63\% vs. 43.96\%). These results highlight both the challenge of transferring from natural to medical video domains and the significant advantage of DISCOVR’s self-supervised representations, especially under limited annotation.
>
> **Table 3. Linear Probing result comparison of models trained on Kinetics-400 large‑scale datasets vs DISCOVR on the EchoDynamic dataset.**
>
> | Model            | Accuracy | Balanced Accuracy | F1 Score |
> |-|:-:|:-:|:-:|
> | ViViT           | 69.76    | 69.75            | 69.74    |
> | TimeSformer      | 73.68    | 73.47            | 73.47    |
> | **DISCOVR (Ours)** | **77.68** | **77.61**        | **77.63** |
>
> ---
>
> **Table 4. Comparison of fully supervised ViViT (trained from scratch) versus self‑supervised DISCOVR (linear probe on the same labeled data).**
>
> | Model                               | Balanced Acc. | Precision | F1‑Score |
> |-|:-:|:-:|:-:|
> | ViViT‑supervised (from scratch)     | 53.45         | 59.50     | 43.96    |
> | DISCOVR (linear probe)         | **77.68**     | **77.61** | **77.63** |
>
> **Q4. Evidence of Generalization to other Modality**
>
> **Answer.**   Most medical imaging modalities, aside from echocardiography, consist primarily of static images or 3D volumes, which results in very limited availability of medical video datasets for research. For example, Cholec80 includes 80 laparoscopic surgery videos, HeiCo has 30 colorectal surgery videos, and EndoVis challenge datasets typically contain fewer than 50 videos per task. Hence, large-scale self-supervised pretraining and direct evaluation of DISCOVR on other medical video datasets are not feasible.
>
> To test whether DISCOVR generalizes beyond echocardiography, we evaluated its transfer performance on two distinct medical image benchmarks: the Breast Ultrasound Images dataset [1] (cancer detection across 600 patients) and DermMNIST [2] (skin lesion classification). Both breast ultrasound and echocardiography require the detection of small, irregular regions of altered tissue, such as hypoechoic tumors in the breast or localized wall motion abnormalities in the heart, making the ability to identify subtle structural changes in one domain directly applicable to the other. Similarly, DermMNIST demands fine-grained visual discrimination between morphologically similar skin lesions. For both benchmarks, we froze the DISCOVR video encoder and trained a linear classifier, comparing performance directly across methods.
>
> As shown in **Table 5**, our method demonstrates strong generalization across both tasks. On the Breast Ultrasound dataset, DISCOVR improves balanced accuracy by 2.01\% over VideoMAE, 19.83\% over SIGMA, and 12.01\% over MGMAE. For DermMNIST, DISCOVR achieves an accuracy of 71.68\%, outperforming VideoMAE by 2.85\%, SIGMA by 3.00\%, and MGMAE by 3.85\%. These results demonstrate strong generalization to multiple medical image analysis tasks beyond echocardiography.
> Further, to assess generalization to natural video data, we pretrained and evaluated all models on the Kinetics 400 action recognition benchmark, using a zero-shot protocol where KNN classification with $K=20$ was applied to features using 64 frames from the frozen video backbone. As shown in **Table 6**, DISCOVR achieves the highest Top-1 accuracy at 22.3\%, outperforming MVD by 3.6\%, MME by 3.2\%, and VideoMAE by 1.6\%, while also requiring the fewest pretraining epochs. These results highlight that DISCOVR not only excels at medical video tasks but also learns generalizable representations efficiently for large-scale natural video datasets.
>
> **Table 5. Linear‑probing results on the Breast Ultrasound dataset and DermMNIST.
> For Breast Ultrasound, we report Balanced Accuracy (Bal. Acc.) and F1; for DermMNIST, overall Accuracy (Acc.).**
>
> | Method | Breast US (Bal. Acc.) | Breast US (F1) | **┃** | DermMNIST (Acc.) |
> |-|:-:|:-:|:-:|:-:|
> | VideoMAE         | 61.75 | 64.45 | **┃** | 68.83 |
> | SIGMA               | 43.93 | 42.21 | **┃** | 68.68 |
> | MGMAE            | 51.75 | 52.34 | **┃** | 67.83 |
> | **DISCOVR (Ours)**  | **63.76** | **65.44** | **┃** | **71.68** |
>
> **Table 6. Zero‑shot KNN classification performance on Kinetics‑400.**
>
> | Model | Epochs | Top‑1 Accuracy (%) |
> |-|:------:|:-:|
> | MVD             | 1600 | 18.7 |
> | MME             |  800 | 19.1 |
> | VideoMAE        |  800 | 20.7 |
> | **DISCOVR (Ours)** | **400** | **22.30** |
>
> ---
>
> We thank the reviewer for their valuable feedback and  insightful questions regarding novelty, scalability, and generalization. We have clarified the distinct contributions of our method and provided new results on computational efficiency and transfer. If any questions remain unresolved, we are happy to provide further clarification or engage in additional discussion.
>
> ---
>
> ## References
>
> [1] Al-Dhabyani, W., Gomaa, M., Khaled, H. and Fahmy, A., 2020. **Dataset of breast ultrasound images.** *Data in brief, 28, p.104863.*
>
> [2] Yang, J., Shi, R., Wei, D., Liu, Z., Zhao, L., Ke, B., Pfister, H. and Ni, B., 2023. **Medmnist v2-a large-scale lightweight benchmark for 2d and 3d biomedical image classification.** *Scientific Data, 10(1), p.41.*

---

> ### Author Response · Authors · 2025-08-05
>
> Dear Reviewer fPeB,
>
> Thank you for your engagement and for confirming receipt of our responses. As we approach the end of the author-reviewer discussion period, please let us know if our clarifications fully address your points, or if you have any unanswered questions before the deadline. We are happy to provide any remaining details.
>
> Best regards,
>
> Authors

---

> > ### Comment · Reviewer_fPeB · 2025-08-05
> >
> > The authors have addressed my concerns. I did not identify other major issues in this paper.

---

> > > ### Author Response · Authors · 2025-08-05
> > >
> > > Dear Reviewer fPeB,
> > >
> > > Thank you for your valuable review and constructive feedback! We are glad to hear that our rebuttal addressed your concerns.
> > >
> > > Sincerely,
> > > Authors

---

### Official Review · Reviewer_Jw9e · 2025-07-03

**Clarity:** 3
**Significance:** 3
**Originality:** 3
**Rating:** 4
**Confidence:** 3

**Summary:**

This paper introduces DISCOVR, a novel self-supervised learning (SSL) framework for echocardiography (cardiac ultrasound) videos. Echocardiography poses unique challenges due to subtle anatomical structures, high frame similarity, and low signal-to-noise ratios. DISCOVR addresses these by combining: Temporal modeling, Spatial semantics, and Cross-modal distillation. The extensive experiments on six datasets show its effectiveness compared to the SOTA SSL and anomaly detection methods.

**Questions:**

How effective of the proposed method in generalizing to other medical imaging modalities except for the echocardiography video?

**Ethical Concerns:**

["NO or VERY MINOR ethics concerns only"]

**Final Justification:**

My final recommendation and evaluation of this paper is 4: Borderline accept.

**Limitations:**

The code is not ready for release yet during the submission and review period. And the reproduction instruction is not provided.

**Quality:**

3

**Strengths And Weaknesses:**

Strengths:

- The research topic on the representation learning of echocardiographic video is meaningful and helpful for real-world applications.
- The paper is well-structured and easy to read with clear formulations.
- The proposed method, combining three loss components, is proven to be effective in the evaluation.
- The experiments are comprehensive, covering diverse datasets and baselines.

Weaknesses:

- Lack of in-depth analysis on the contribution of each component to the final performance. For example, how do the encoded images help the video modality training? Some visualizations or case studies, e.g., the separation of hidden state features before and after adding the image modality, may help.
- The ablation studies on the effect of components (i.e., Table 4) are insufficient to fully prove the effectiveness of each component of the proposed method. For example, the image SSL loss is ignored.
- The direct addition of two modalities in Equation (9) lacks a convincing explanation, since normally, there exists a balance coefficient to test the best performance.

---

> ### Author Rebuttal · Authors · 2025-07-30
>
> **Q1.** **Include image-SSL baseline and visualizations to understand the effect of image-guidance**
>
> **Answer.** To address the reviewer’s concern regarding the contribution of each modality in our framework, we have included explicit baselines in **Table 1** that isolate the performance of models using only image or only video self-distillation. Our results show that neither video-only nor image-only models achieve F1 scores above 50\%, underscoring that neither modality alone is sufficient for strong video representation learning. In contrast, combining both modalities with our cross-modal SCD loss yields a significant improvement, resulting in 61.45\% F1 and 63.20\% balanced accuracy. This demonstrates that the integration of spatial and temporal encoding is essential and that our SCD mechanism is crucial for unifying these features to produce robust and clinically meaningful representations. Additionally, we have generated self-attention heatmaps (in the style of Figure 3) that qualitatively compare models with and without the image modality. These visualizations show that the inclusion of the image encoder helps the model focus more effectively on fine-grained, clinically relevant structures. Due to NeurIPS policy, we cannot include these figures at this stage, but we will make them available in the camera-ready version to provide further qualitative support for our method.
>
> **Table 1. Effect of the different loss terms on classification performance (Balanced Accuracy, Precision, and F1).**
>
> | $\mathcal{L}^{vid}_{\text{ssl}}$ | $\mathcal{L}^{img}_{\text{ssl}}$ | $\mathcal{L}_{\text{SCD}}$ | Bal. Acc. | Precision | F1 |
> |:--------------------------------:|:--------------------------------:|:-------------------------:|:---------:|:---------:|:--:|
> | **✓** | **✗** | **✗** | 52.27 | 53.16 | 48.23 |
> | **✗** | **✓** | **✗** | 53.66 | 55.22 | 49.43 |
> | **✓** | **✓** | **✓** | **63.20** | **65.35** | **61.45** |
>
> **Q2. Ablation of Loss Weight Terms**
>
> **Answer.** We agree with the reviewer about the potential importance of balancing the loss terms. In our experiments, simply combining the loss terms with equal weighting already yielded strong, state-of-the-art (SOTA) results. To further assess the effect of different weightings, we trained models for 200 epochs (as training a full model takes a long time and is computationally expensive) using several alternative loss weight combinations as shown in **Table 2**. While the differences in performance across these settings were not substantial, the configuration $\lambda_{\text{img}} = 0.8$ and $\lambda_{\text{vid}} = 0.2$ gave the best overall results, achieving 56.28\% Accuracy, 56.27\% Balanced Accuracy, and an F1 score of 56.14\%. We thank the reviewer for the suggestion that led to increase in performance. We will include the complete analysis by training for the full 400 epochs with more combinations, as well as additional details, in the camera-ready version.
>
> **Table 2. Effect of loss weights ($\lambda_{\text{img}}$ for the image branch and $\lambda_{\text{vid}}$ for the video branch) on linear‑probing classification performance on the EchoNet‑Dynamic dataset.**
>
> | $\lambda_{\text{img}}$ | $\lambda_{\text{vid}}$ | Accuracy | Balanced Accuracy | F1 Score |
> |:----------------------:|:----------------------:|:--------:|:-----------------:|:--------:|
> | 1.0 | 1.0 | 55.52 | 55.51 | 55.47 |
> | 0.2 | 0.8 | 55.04 | 55.05 | 54.84 |
> | 0.2 | 0.5 | 55.80 | 55.76 | 54.64 |
> | 0.8 | 0.2 | **56.28** | **56.27** | **56.14** |
> | 0.5 | 0.2 | 55.60 | 55.57 | 55.02 |
>
> **Q3. Generalization to Other Modalities.**
>
> **Answer.**   Most medical imaging modalities, aside from echocardiography, consist primarily of static images or 3D volumes, which results in very limited availability of medical video datasets for research. For example, Cholec80 includes 80 laparoscopic surgery videos, HeiCo has 30 colorectal surgery videos, and EndoVis challenge datasets typically contain fewer than 50 videos per task. Hence, large-scale self-supervised pretraining and direct evaluation of DISCOVR on other medical video datasets are not feasible.
>
> To test whether DISCOVR generalizes beyond echocardiography, we evaluated its transfer performance on two distinct medical image benchmarks: the Breast Ultrasound Images dataset [1] (cancer detection across 600 patients) and DermMNIST [2] (skin lesion classification). Both breast ultrasound and echocardiography require the detection of small, irregular regions of altered tissue, such as hypoechoic tumors in the breast or localized wall motion abnormalities in the heart, making the ability to identify subtle structural changes in one domain directly applicable to the other. Similarly, DermMNIST demands fine-grained visual discrimination between morphologically similar skin lesions. For both benchmarks, we froze the DISCOVR video encoder and trained a linear classifier, comparing performance directly across methods.
>
> As shown in **Table 3**, our method demonstrates strong generalization across both tasks. On the Breast Ultrasound dataset, DISCOVR improves balanced accuracy by 2.01\% over VideoMAE, 19.83\% over SIGMA, and 12.01\% over MGMAE. For DermMNIST, DISCOVR achieves an accuracy of 71.68\%, outperforming VideoMAE by 2.85\%, SIGMA by 3.00\%, and MGMAE by 3.85\%. These results demonstrate strong generalization to multiple medical image analysis tasks beyond echocardiography.
>
> Further, to assess generalization to natural video data, we pretrained on 16 frame clips (default for the benchmark) and evaluated all models on the Kinetics 400 action recognition benchmark, using a zero-shot protocol where KNN classification with $K=20$ was applied to features using 16 frames  from the frozen video backbone. As shown in **Table 4**, DISCOVR achieves the highest Top-1 accuracy at 22.3\%, outperforming MVD by 3.6\%, MME by 3.2\%, and VideoMAE by 1.6\%, while also requiring the fewest pretraining epochs. These results highlight that DISCOVR not only excels at medical video tasks but also learns generalizable representations efficiently for large-scale natural video datasets.
>
> **Table 3. Linear‑probing results on the Breast Ultrasound dataset and DermMNIST.
> For Breast Ultrasound, we report Balanced Accuracy (Bal. Acc.) and F1; for DermMNIST, overall Accuracy (Acc.).**
>
> | Method | Breast US (Bal. Acc.) | Breast US (F1) | **┃** | DermMNIST (Acc.) |
> |--------|:--------------------:|:--------------:|:----:|:---------------:|
> | VideoMAE [3]        | 61.75 | 64.45 | **┃** | 68.83 |
> | SIGMA [4]              | 43.93 | 42.21 | **┃** | 68.68 |
> | MGMAE [5]           | 51.75 | 52.34 | **┃** | 67.83 |
> | **DISCOVR (Ours)**  | **63.76** | **65.44** | **┃** | **71.68** |
>
> **Table 4. Zero‑shot KNN classification performance on Kinetics‑400.**
>
> | Model | Epochs | Top‑1 Accuracy (%) |
> |-------|:------:|:------------------:|
> | MVD [6]            | 1600 | 18.7 |
> | MME [7]            |  800 | 19.1 |
> | VideoMAE [3]       |  800 | 20.7 |
> | **DISCOVR (Ours)** | **400** | **22.30** |
>
>
> **Q4. Code Release and Reproducibility**
>
> **Answer.** We appreciate the reviewer’s concern regarding reproducibility. We have already included our code in the supplementary materials for inspection. Additionally, we have released code and all necessary instructions for reproduction on anonymous GitHub repository but we are unable to share the repository link at this stage due to NeurIPS guidelines. Full access to the repository and detailed instructions will be provided in the camera-ready version to ensure easy reproducibility for the community.
>
> ---
>
> We thank the reviewer for their constructive feedback and specific suggestions regarding the image-SSL baseline, loss weighting, generalization to other modalities, and reproducibility. Your input prompted us to add explicit ablation baselines, experiment with alternative loss weightings, and expand our evaluation to external medical datasets and natural video benchmarks. We have also clarified our code release plans. If any questions remain unresolved, we are happy to provide further clarification or engage in additional discussion.
>
> ---
>
> ## References
> [1] Al-Dhabyani, W., Gomaa, M., Khaled, H. and Fahmy, A., 2020. **Dataset of breast ultrasound images.** *Data in brief, 28, p.104863.*
>
> [2] Yang, J., Shi, R., Wei, D., Liu, Z., Zhao, L., Ke, B., Pfister, H. and Ni, B., 2023. **Medmnist v2-a large-scale lightweight benchmark for 2d and 3d biomedical image classification.** *Scientific Data, 10(1), p.41.*
>
> [3] Tong, Z., Song, Y., Wang, J. and Wang, L., 2022. Videomae: **Masked autoencoders are data-efficient learners for self-supervised video pre-training.** *Advances in neural information processing systems, 35, pp.10078-10093.*
>
> [4]  Salehi, M., Dorkenwald, M., Thoker, F.M., Gavves, E., Snoek, C.G. and Asano, Y.M., 2024, September. **Sigma: Sinkhorn-guided masked video modeling.** *In European Conference on Computer Vision (pp. 293-312). Cham: Springer Nature Switzerland.*
>
> [5] Huang, B., Zhao, Z., Zhang, G., Qiao, Y. and Wang, L., 2023. **Mgmae: Motion guided masking for video masked autoencoding.** *In Proceedings of the IEEE/CVF International Conference on Computer Vision (pp. 13493-13504).*
>
> [6] Wang, R., Chen, D., Wu, Z., Chen, Y., Dai, X., Liu, M., Yuan, L. and Jiang, Y.G., 2023. **Masked video distillation: Rethinking masked feature modeling for self-supervised video representation learning.** *In Proceedings of the IEEE/CVF conference on computer vision and pattern recognition (pp. 6312-6322).*
>
> [7] Sun, X., Chen, P., Chen, L., Li, C., Li, T.H., Tan, M. and Gan, C., 2023. **Masked motion encoding for self-supervised video representation learning.** *In Proceedings of the IEEE/CVF conference on computer vision and pattern recognition (pp. 2235-2245).*

---

> > ### Comment · Reviewer_Jw9e · 2025-08-05
> > **I thank the authors for their rebuttal. I have no further questions.**
> >
> > I thank the authors for their rebuttal. I have no further questions.

---

> > > ### Author Response · Authors · 2025-08-05
> > >
> > > Thank you for your valuable review and constructive feedback! We are glad to hear that our rebuttal addressed your concerns.
> > >
> > >
> > > Sincerely,
> > >
> > > Authors

---

### Official Review · Reviewer_6qum · 2025-07-05

**Clarity:** 3
**Significance:** 3
**Originality:** 3
**Rating:** 4
**Confidence:** 3

**Summary:**

This work introduces DISCOVR, a self-supervised learning framework for echocardiography video representation. It combines masked video self-distillation, masked image self-distillation, and image-to-video semantic cluster distillation to capture both temporal dynamics and fine-grained spatial semantics. Experiments on six echocardiography datasets demonstrate that it outperforms existing SSL methods and anomaly detection methods in tasks like anomaly detection, classification, and segmentation.

**Questions:**

See weaknesses.

**Ethical Concerns:**

["NO or VERY MINOR ethics concerns only"]

**Final Justification:**

Most of my concerns have been addressed. However, the contribution remains incremental in terms of novelty. Therefore, I keep my original score.

**Limitations:**

No.

The authors briefly mention in the appendix that their study focuses on echocardiography, but the discussion of limitations and potential negative societal impacts remains insufficient. A more thorough analysis—particularly one based on failure cases—would help to identify and communicate the boundaries of the method’s applicability. For example, understanding when and why the model fails could reveal biases, generalization limitations, or risks of misuse in clinical settings. I encourage the authors to include a more detailed discussion of these aspects to strengthen the ethical and practical reflection of their work.

**Paper Formatting Concerns:**

There are no paper formatting issues.

**Quality:**

3

**Strengths And Weaknesses:**

Strengths:
1. The paper is clearly written and well-structured.
2. Experimental results are positive and fit the paper.

Weaknesses:
1. In the ablation study, to rigorously evaluate the contribution of the video self-distillation component, a baseline that includes only the masked image self-distillation (without the video component) should be provided. This comparison is necessary to isolate and quantify the effect of video self-distillation.
2. Left Ventricular Ejection Fraction (LVEF) is a critical clinical metric. However, the paper only evaluates performance on a binary classification task (normal vs. abnormal). It would strengthen the contribution if the model could also be evaluated on a regression task—i.e., directly predicting the LVEF score.
3. The method for generating the heatmap shown in Figure 3 is not clearly described. More details are needed to understand how it was computed, including what features or gradients were used and how the visualization reflects model interpretability.

---

> ### Author Rebuttal · Authors · 2025-07-30
>
> **Q1**. **Add Baseline for image self-distillation to ablation study.**
>
> **Answer.**  Thank you for your suggestion. To rigorously evaluate each component, we have added the baseline using only masked image  self-distillation to the loss ablation table (Table 4a). Indeed, we find that the image and video self distillation perform suboptimally in isolation, as shown in **Table 1**, confirming that spatial or temporal cues alone are insufficient for strong representation learning. In contrast, combining both with the SCD loss, which explicitly distills fine-grained semantic structure from the image branch into the video backbone, achieves the best results. This supports our intuition that SCD is crucial for aligning spatial semantics with temporal dynamics, enabling more robust and clinically meaningful video representations. We will add this to our ablation tables in the camera-ready version.
>
> **Table 1. Effect of the different loss terms on classification performance (Balanced Accuracy, Precision, and F1).**
>
>
> | $\mathcal{L}^{vid}_{\text{ssl}}$ | $\mathcal{L}^{img}_{\text{ssl}}$ | $\mathcal{L}_{\text{SCD}}$ | Bal. Acc. | Precision | F1 |
> |:--------------------------------:|:--------------------------------:|:-------------------------:|:---------:|:---------:|:--:|
> | **✓** | **✗** | **✗** | 52.27 | 53.16 | 48.23 |
> | **✗** | **✓** | **✗** | 53.66 | 55.22 | 49.43 |
> | **✓** | **✓** | **✓** | **63.20** | **65.35** | **61.45** |
>
> **Q2.** **Include Ejection Fraction downstream task**
>
> **Answer.** We thank the reviewer for the valuable suggestion to evaluate LVEF regression. We performed this analysis, and the results are presented below and in **Table 2**.
>
> First, to establish a direct comparison with other SSL baselines, we adopted a standard linear probing setup, training only a linear layer on top of frozen self-supervised features. As shown in Table 2, our DISCOVR model achieves a Mean Absolute Error (MAE) of 7.79, demonstrating superior performance over the baselines.
>
> To further evaluate the strength of DISCOVR, we then finetuned only the last three blocks of our encoder. This significantly boosts performance, reducing the MAE to 6.32. This result is particularly compelling when compared to the state-of-the-art fully supervised model, EchoNet-Dynamic [1]. The multi-modal EchoNet-Dynamic model achieves an MAE of 4.05 by leveraging a large, multi-task architecture (ejection fraction regression + segmentation) of 71.1 million parameters that is co-trained with an extensive set of 20,060 manually segmented labels.
>
> For a more direct and fair comparison to our single-task setup (LVEF prediction), we benchmarked our model against the EchoNet-Dynamic base model and other supervised baselines from [1] which are trained only on EF data. In this comparison, our self-supervised approach demonstrates a clear advantage. Our partially fine-tuned DISCOVR model (MAE 6.32) outperforms these fully-supervised counterparts, including MC3 (MAE 6.59), the base EchoNet-Dynamic model (MAE 7.35), and R3D (MAE 7.63). This demonstrates DISCOVR's learned representations effectively generalize to clinically relevant regression tasks such as ejection fraction (EF) prediction.
>
> **Table 2. LVEF prediction results on the EchoNet‑Dynamic dataset, detailing supervision type. Our self‑supervised method is compared against other SSL methods and fully‑supervised baselines from [1].**
>
> | Model | MAE&nbsp;↓ | RMSE&nbsp;↓ | EF Labels | Seg. Labels |
> |-------|:---------:|:-----------:|:---------:|:-----------:|
> | **Self‑Supervised (Linear Probing)** |||||
> | VideoMAE | 8.02 | 11.16 | ✓ |   |
> | MGMAE | 8.88 | 12.47 | ✓ |   |
> | DISCOVR (Ours) | **7.79** | **10.89** | ✓ |   |
> | **Fully‑Supervised Baselines [1] (trained only with EF data)** |||||
> | MC3 (All frames) | 6.59 | 9.39 | ✓ |   |
> | EchoNet‑Dynamic (All frames) | 7.35 | 9.53 | ✓ |   |
> | R3D (All frames) | 7.63 | 9.75 | ✓ |   |
> | DISCOVR (finetune last 3 blocks, 64 frames) | **6.32** | **8.62** | ✓ |   |
> | **Supervised with Segmentation & EF labels** |||||
> | EchoNet‑Dynamic (Full model) | **4.05** | **5.30** | ✓ | ✓ |
>
>
> **Q3.** **How were visualisations in Figure 3 generated?**
>
> **Answer**. For Fig. 3, we use a self-attention map visualization method, similar to that of  DINO [2], to evaluate what each model attends to within the input video.  This technique extracts self-attention scores from the final layer of our video-encoder and projects them onto the video frames, providing a direct illustration of “what the model is looking at” as it forms its internal representation. The approach relies solely on the model’s native attention weights, without the use of gradients or class-specific features. This visualization highlights the spatial and temporal regions that are most influential for the model’s reasoning. In the case of our model, these regions correspond to clinically relevant cardiac structures and their motion throughout the heart cycle, supporting the interpretability and clinical utility of the learned representations. We will add the explanation to the figure's caption in the final version.
>
> **Q4.** **Discussion regarding societal impact and limitations.**
>
> **Answer.**
> We appreciate this important feedback. In the broader impact section present in the appendix, we discuss that our method has the potential to assist sonographers in screening by flagging subtle abnormalities in echocardiography scans. However, as also noted, DISCOVR requires rigorous clinical validation before it can be used in clinical practice. This validation is essential for addressing the significant risks of inherent biases that could otherwise lead to unreliable or inequitable outcomes. For example, models must be assessed for population and demographic bias, where performance can differ across patient demographics [3,4], as well as machine and algorithmic bias, where a model may learn from site-specific imaging artifacts rather than true pathology [5,6]. Furthermore, AI can get biased towards the subjective and sometimes flawed cognitive biases of human experts, a form of operator and annotation bias that gets embedded in the training data [7]. Finally, issues like prevalence bias, where a model's predictive accuracy is skewed by the frequency of a disease in its training data, must also be carefully evaluated.
>
> To this end, we are currently conducting prospective evaluations across five clinical sites in diverse geographical regions, with a particular focus on systematically identifying and analyzing model failure cases, evaluating performance across patient subgroups and specific congenital heart disease (CHD) conditions, and defining the boundaries of generalization. The outcome of this ongoing work, including limitations and potential risks relevant to clinical deployment, will be carefully documented and published. We will add a detailed description regarding the same in the camera-ready version.
>
> ---
>
> We thank the reviewer for their thoughtful comments and constructive suggestions. Your requests regarding the image self-distillation ablation, the LVEF regression task, and detailed evaluation metrics have led us to  expand our experiments and clarify both our implementation and broader implications. If any questions remain unresolved, we are happy to provide further clarification or engage in additional discussion.
>
> ---
> ## References
>
> [1] Ouyang, D., He, B., Ghorbani, A., Yuan, N., Ebinger, J., Langlotz, C.P., Heidenreich, P.A., Harrington, R.A., Liang, D.H., Ashley, E.A. and Zou, J.Y., 2020. **Video-based AI for beat-to-beat assessment of cardiac function.** *Nature, 580(7802), pp.252-256.*
>
> [2] Caron, M., Touvron, H., Misra, I., Jégou, H., Mairal, J., Bojanowski, P. and Joulin, A., 2021. **Emerging properties in self-supervised vision transformers.** *In Proceedings of the IEEE/CVF international conference on computer vision (pp. 9650-9660).*
>
> [3] Liu, Y., Jain, A., Eng, C., Way, D.H., Lee, K., Bui, P., Kanada, K., de Oliveira Marinho, G., Gallegos, J., Gabriele, S. and Gupta, V., 2020. **A deep learning system for differential diagnosis of skin diseases**. *Nature medicine, 26(6), pp.900-908.*
>
> [4] Gichoya, J.W., Banerjee, I., Bhimireddy, A.R., Burns, J.L., Celi, L.A., Chen, L.C., Correa, R., Dullerud, N., Ghassemi, M., Huang, S.C. and Kuo, P.C., 2022. **AI recognition of patient race in medical imaging: a modelling study.** *The Lancet Digital Health, 4(6), pp.e406-e414.*
>
> [5] Zech, J.R., Badgeley, M.A., Liu, M., Costa, A.B., Titano, J.J. and Oermann, E.K., 2018. **Variable generalization performance of a deep learning model to detect pneumonia in chest radiographs: a cross-sectional study.** *PLoS medicine, 15(11), p.e1002683.*
>
> [6] Howard, F.M., Dolezal, J., Kochanny, S., Schulte, J., Chen, H., Heij, L., Huo, D., Nanda, R., Olopade, O.I., Kather, J.N. and Cipriani, N., 2021. **The impact of site-specific digital histology signatures on deep learning model accuracy and bias.** *Nature communications, 12(1), p.4423.*
>
> [7] Vrudhula, A., Kwan, A.C., Ouyang, D. and Cheng, S., 2024. **Machine learning and bias in medical imaging: opportunities and challenges.** *Circulation: Cardiovascular Imaging, 17(2), p.e015495.*

---

> ### Author Response · Authors · 2025-08-05
>
> Dear Reviewer 6qum,
>
> Thank you for your thorough review. Since the author-reviewer discussion period is nearing its end, could you please let us know if our rebuttal resolves your concerns or if further discussion would be helpful? We are happy to address any remaining questions you may have.
>
> Best regards,
>  Authors

---

> > ### Comment · Reviewer_6qum · 2025-08-05
> >
> > Thanks for the authors' response. Most of my concerns have been addressed. I have no further questions.

---

> > > ### Author Response · Authors · 2025-08-05
> > >
> > > Dear Reviewer  6qum,
> > >
> > > Thank you for your valuable review and constructive feedback! We are glad to hear that our rebuttal addressed your concerns.
> > >
> > > Sincerely,
> > > Authors

---

### Note · Authors · 2025-08-12

Dear Reviewers (and AC), thank you for your time and the constructive discussion. The exchange resulted in substantial improvements to the paper and resolution of key concerns. We only provide an overview here for convenience:

### Updates & Upgrades
- **Generalization & Clinical Relevance** – Added transfer evaluations to Breast Ultrasound, DermMNIST, and Kinetics-400, confirming that DISCOVR learns generalizable representations beyond echocardiography. Extended downstream evaluation to LVEF regression, achieving superior performance over fully supervised EF-only baselines (e.g., MC3, EchoNet-Dynamic base, R3D).
- **Interpretability & Broader Impact** – Detailed the self-attention visualization method (DINO-style attention maps highlighting clinically meaningful structures) and expanded the societal-impact/limitations section; initiated prospective evaluations across five clinical sites to assess bias and define generalization boundaries.
- **Novelty & Methodological Clarity** – Further clarified the originality of our cross-modal Semantic Cluster Distillation (SCD) loss and explained how it unifies spatial and temporal representations, distinguishing it from prior SSL methods.
- **Expanded Ablations** – Introduced explicit image-only and video-only baselines, along with experiments varying loss weights. Results confirmed that neither modality alone is sufficient and that SCD is essential for robust representation learning.
- **Efficiency & Scalability** – Provided computational-cost and scalability analysis, showing that improvements incur minimal additional training overhead and no inference penalty.
- **Fairness of Comparisons** – Verified identical protocols for all baselines, justified our tokenization and sampling strategy, and improved clarity in the mathematical presentation.

### Reviewer Stated Outcomes
- **jDmx** — Explicitly raised their score to 4 after the rebuttal, noting that the revisions addressed their concerns.
- **6qum** — Confirmed that all main concerns were resolved and no further questions remained.
- **Jw9e** — Expressed satisfaction with the added baselines, cross-domain transfer results, and reproducibility plans.
- **fPeB** — Agreed that the revisions addressed their comments and left no remaining issues.

---

### Decision · Program_Chairs · 2025-09-17

**Decision:**

Accept (poster)

**Comment:**

This paper received all positive reviews. After discussion, reviewers agreed to accept this paper.  The authors are encouraged to address the major concerns in the camera-ready version.
1. Add experimental evaluation on regression tasks.
2. Add ablation studies to evaluate each component in the model
3. Add results on computational cost

The authors’ rebuttal and messages were carefully ready, discussed, and considered.